# MiLoPYP: self-supervised molecular pattern mining and particle localization in situ

Qinwen Huang ®[1,4], Ye Zhou ®[1,4] & Alberto Bartesaghi ®[1,2,3] ✉

Cryo-electron tomography allows the routine visualization of cellular landscapes in three dimensions at nanometer-range resolutions. When combined with single-particle tomography, it is possible to obtain near-atomic resolution structures of frequently occurring macromolecules within their native environment. Two outstanding challenges associated with cryo-electron tomography/single-particle tomography are the automatic identification and localization of proteins, tasks that are hindered by the molecular crowding inside cells, imaging distortions characteristic of cryo-electron tomography tomograms and the sheer size of tomographic datasets. Current methods suffer from low accuracy, demand extensive and time-consuming manual labeling or are limited to the detection of specific types of proteins. Here, we present MiLoPYP, a two-step dataset-specific contrastive learning-based framework that enables fast molecular pattern mining followed by accurate protein localization. MiLoPYP's ability to effectively detect and localize a wide range of targets including globular and tubular complexes as well as large membrane proteins, will contribute to streamline and broaden the applicability of high-resolution workflows for in situ structure determination.

Cryo-electron tomography (CET) has recently emerged as the leading imaging technology for visualizing intricate cellular landscapes at nanometer resolution[1–6]. By taking a sequence of tilted projections of the sample, CET allows structural studies of macromolecular complexes imaged within their native context providing valuable insights into their organization and interactions with partner molecules. To analyze these complexes effectively, it is crucial to determine the composition of the cellular environment and to precisely localize molecular species within the cellular milieu. This is needed for subsequent single-particle tomography (SPT) analysis, which allows the determination of higher-resolution structures from repeating or frequently occurring targets within the sample[7,8]. The complexity and crowdedness of cellular environments compounded with the low signal-to-noise ratios and the image distortions caused by the missing wedge, however, pose substantial technical challenges[2,9]. Moreover, these problems are exacerbated by the large volumes of data that are routinely produced using high-throughput strategies for sample preparation[10–12], autonomous grid navigation[13,14] and multi-shot tomography[15–17].

Current solutions to the problem of cellular pattern mining include template matching[18–21] and manual or semiautomatic approaches that require substantial amounts of user intervention[22,23]. Template-based methods are prone to model bias and are computationally intensive, especially when dealing with large datasets and multiple protein species. Inspired by the success of methods based on convolutional neural networks (CNNs) to pick particles from two-dimensional (2D) micrographs[24–29], CNN-based approaches were introduced to pick particles from 3D tomograms[28,30–32]. These fully supervised methods, however, require extensive data annotation and generalize poorly to previously unseen datasets. More recently, DISCA, an unsupervised clustering-based approach was introduced[33] that learns to classify subtomograms by modeling data distributions using expectation–maximization. However, DISCA suffers from training instability,

[1]Department of Computer Science, Duke University, Durham, NC, USA. [2]Department of Biochemistry, Duke University School of Medicine, Durham, NC, USA. [3]Department of Electrical and Computer Engineering, Duke University, Durham, NC, USA. [4]These authors contributed equally: Qinwen Huang, Ye Zhou. ✉e-mail: alberto.bartesaghi@duke.edu

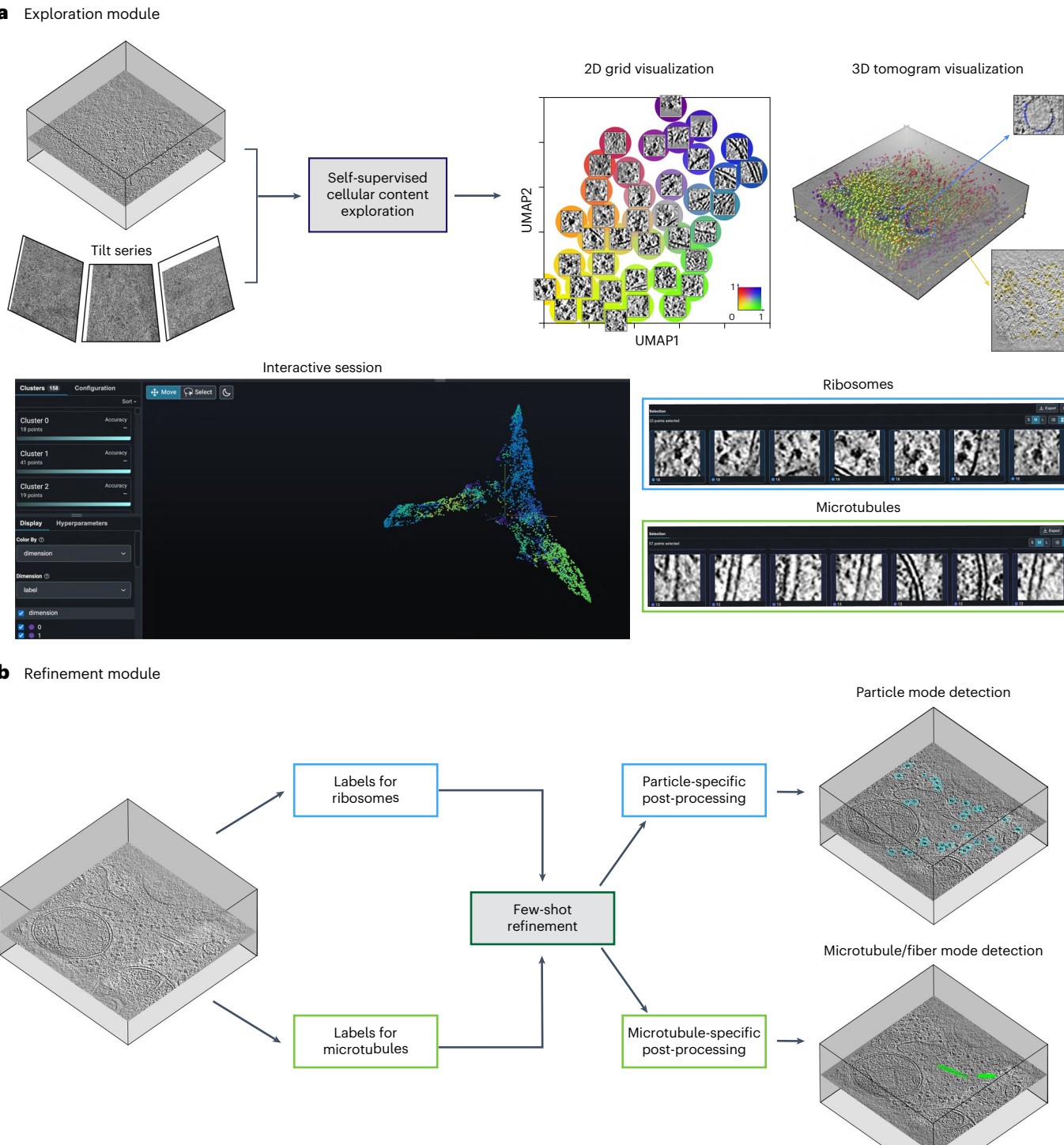

**a** Exploration module

2D grid visualization

3D tomogram visualization

Interactive session

Ribosomes

Microtubules

**b** Refinement module

Particle mode detection

Microtubule/fiber mode detection

**Fig. 1 | Overview of MiLoPYP modules for self-supervised cellular content exploration and semi-supervised particle localization. a**, The cellular content exploration module learns an embedding for each subtomogram and corresponding patches from the aligned tilt series. The embeddings can be visualized in three different ways: (1) the 2D grid visualization displays patches in a 0–1 grid based on the learned embedding coordinates using a rainbow color scheme (top, middle); (2) the 3D tomogram visualization maps patches back to the 3D tomogram coloring the corresponding center coordinates based on the embedding coordinates (top, right); and (3) an interactive session displays embeddings in 3D space allowing the user to perform interactive selection, visualization and export of training coordinates for specific protein species (bottom). **b**, MiLoPYP's refinement module performs protein localization using few-shot learning followed by a protein-specific post-processing step that determines particle locations based on their shape. Training is done using partially annotated tomograms produced by the cellular content exploration module. The final output consists of 3D tomogram coordinates for each particle species that can be used for downstream SPT.

assumes a discrete number of clusters and requires that large continuous structures such as membranes be separately annotated, thus increasing the amount of user intervention. To partially mitigate these problems, a representation learning-based approach called TomoTwin was proposed[34]. TomoTwin learns to map each voxel and its surrounding content into high-dimensional subspace and the relative location of macromolecules is determined by measuring similarities between the embeddings. Still, this fully supervised approach requires extensive training using tens of thousands of simulated tomograms of known proteins obtained from the Protein Data Bank (PDB)[35], and has been shown to have limited success on non-globular and unseen proteins. In addition, the use of three-dimensional (3D) subtomograms during training and evaluation makes existing approaches computationally intensive and prone to distortions caused by the missing wedge.

To overcome these challenges, we present MiLoPYP, a dataset-specific cellular pattern Mining (Mi) and particle Localization (Lo) PYthon (PY) Pipeline (P). MiLoPYP operates in two phases: a cellular content exploration step followed by protein-specific particle localization. During the first step, the framework learns to embed subtomograms into a high-dimensional representation in a self-supervised fashion using contrastive representation learning[36–40]. MiLoPYP effectively maps subtomograms into an embedding space where similar data points are positioned close to each other while dissimilar ones are pushed apart, thus improving separability and allowing users to identify frequently occurring proteins. During the second step, MiLoPYP learns to identify specific proteins across entire datasets using few-shot particle detection that only requires sparse labels for training and is computationally efficient[41]. Training data for the protein localization step are obtained by interactively selecting a subset of points from the output of the cellular content exploration step, thus eliminating the need for time-consuming manual labeling. To minimize the effects of the missing wedge and reduce computational costs, we use averages of 2D projections from the raw tilt series and averages of consecutive tomographic slices (instead of 3D subtomograms)[42]. In addition to detecting globular-shaped targets, MiLoPYP can accurately identify membrane-attached proteins and fiber-like proteins, providing a versatile tool for comprehensive cellular content exploration and protein localization that runs fast and requires minimal human intervention.

## Results

### Overview of MiLoPYP's workflow
MiLoPYP is a deep-learning framework comprising a cellular content mining and exploration module followed by protein-specific particle localization (Fig. 1). Both modules require minimal supervision, thereby enhancing their practicality. In the cellular content exploration module, instead of using a naive sliding-window approach centered around each voxel, MiLoPYP utilizes a difference of Gaussian (DoG) pyramid to identify crucial coordinates of interest[43], thus resulting in improved computational efficiency (Extended Data Fig. 1). Sub-volumes centered at these coordinates are then extracted from the tomogram and fed into a Siamese network used for representation learning[40]. By leveraging pairs of augmented sub-volumes as inputs, the network maximizes the similarity between each sub-volume and its augmented counterpart, eliminating the need for ground-truth labels. Once trained, the network effectively learns to group together proteins with similar shapes while assigning proteins with different shapes to distant representations. MiLoPYP offers three methods to visualize the learned embeddings: 2D grid visualization, 3D tomogram visualization and 3D embedding interactive session (Fig. 1a). For 2D grid visualization, 2D feature vectors are assigned to individual *xy* slices of subtomograms and positioned on a 2D grid colored according to normalized feature-coordinate values. For 3D tomogram visualization, MiLoPYP allows mapping the structural diversity present in a dataset by assigning a distinct color to each voxel in the tomogram based on its normalized 2D representation, with similar color voxels representing structurally homogeneous

features. For the 3D embedding interactive session, embeddings are first assigned discrete labels using an over-clustering algorithm and colored according to their embedding coordinates (Methods). Users can then interactively select specific regions of the embedding space and conveniently visualize patches by mapping them to their original tomogram positions. MiLoPYP's visualization tools allow users to conveniently explore and select subsets of frequently occurring particles across large sets of tomograms and use them as input to the protein localization module (Fig. 1b). Since the original DoG-based coordinates typically have low precision, a refinement step is necessary to improve the accuracy of protein localization. MiLoPYP's refinement step is trained in a semi-supervised manner and generates a probability heat map that represents the likelihood of a given protein being present at each voxel in the tomogram. Non-maximum suppression (NMS) is then applied to this probability heat map, followed by post-processing and thresholding using a user-defined probability value (Extended Data Fig. 2). The positions resulting from the refinement step constitute the final 3D coordinate outputs that are used for subsequent SPT refinement.

### Accurate detection and localization of proteins imaged in vitro
We first assessed the performance of MiLoPYP on tilt series from purified *Escherichia coli* 70S ribosomes (EMPIAR-10304)[44]. The feature mapping obtained using the exploration module served as confirmation of the inherent species homogeneity in this dataset. The 2D grid visualization reveals distinct clusters representing 70S ribosomes, gold fiducials and the presence of background elements and contamination (Fig. 2a). The 3D tomogram visualization provides a visual perspective, showing the location of different structural elements within each tomogram. The distance between representations for ribosomes and gold beads is closer in comparison to distances observed between representations for ribosomes and background (or gold fiducials and background), highlighting the discriminatory capacity of MiLoPYP. Using the 3D interactive session, we obtained a subset of particles that shared the same label (assigned through over-clustering). Within this subset, a cluster comprising 230 particles was selected by visual inspection of the cluster centroids (Methods). The particles and their corresponding coordinates from this cluster were used to train the refinement module (Fig. 2b). Following 3D refinement and reconstruction in nextPYP[45], we obtained a 5.0 Å resolution map of the 70S ribosome (Fig. 2c). To assess the robustness of MiLoPYP to the number of annotations, we measured the quality of particle picking using F1 scores as a function of the number of labels (Extended Data Fig. 3a). While having more annotations always improved performance, using only 55 annotations produced a high detection F1 score of 0.73, demonstrating MiLoPYP's ability to locate targets at progressively lower concentrations. We also benchmarked MiLoPYP against conventional template matching[18,46,47] and deep-learning-based approaches crYOLO-3D[28], DeepFinder[31], DeePiCt[32] and TomoTwin[34] (Methods). Our experiments show that MiLoPYP consistently outperforms existing algorithms on challenging Electron Microscopy Public Image Archive (EMPIAR) datasets both in terms of speed and precision/recall scores while requiring minimal user input.

### Detection of molecular patterns and particle identification in situ
Next, we assessed MiLoPYP's ability to map proteins within crowded cellular environments using tomograms from *Mycoplasma pneumoniae* bacterial cells (EMPIAR-10499)[48]. The 2D grid visualization plots revealed the presence of cell membranes as well as 70S ribosomes (Fig. 3a). They also uncovered the presence of smaller-sized particles and vesicles, which we are currently unable to identify due to their limited copy number in the dataset (Extended Data Fig. 4). Using the interactive 3D embedding session, we selected a cluster corresponding

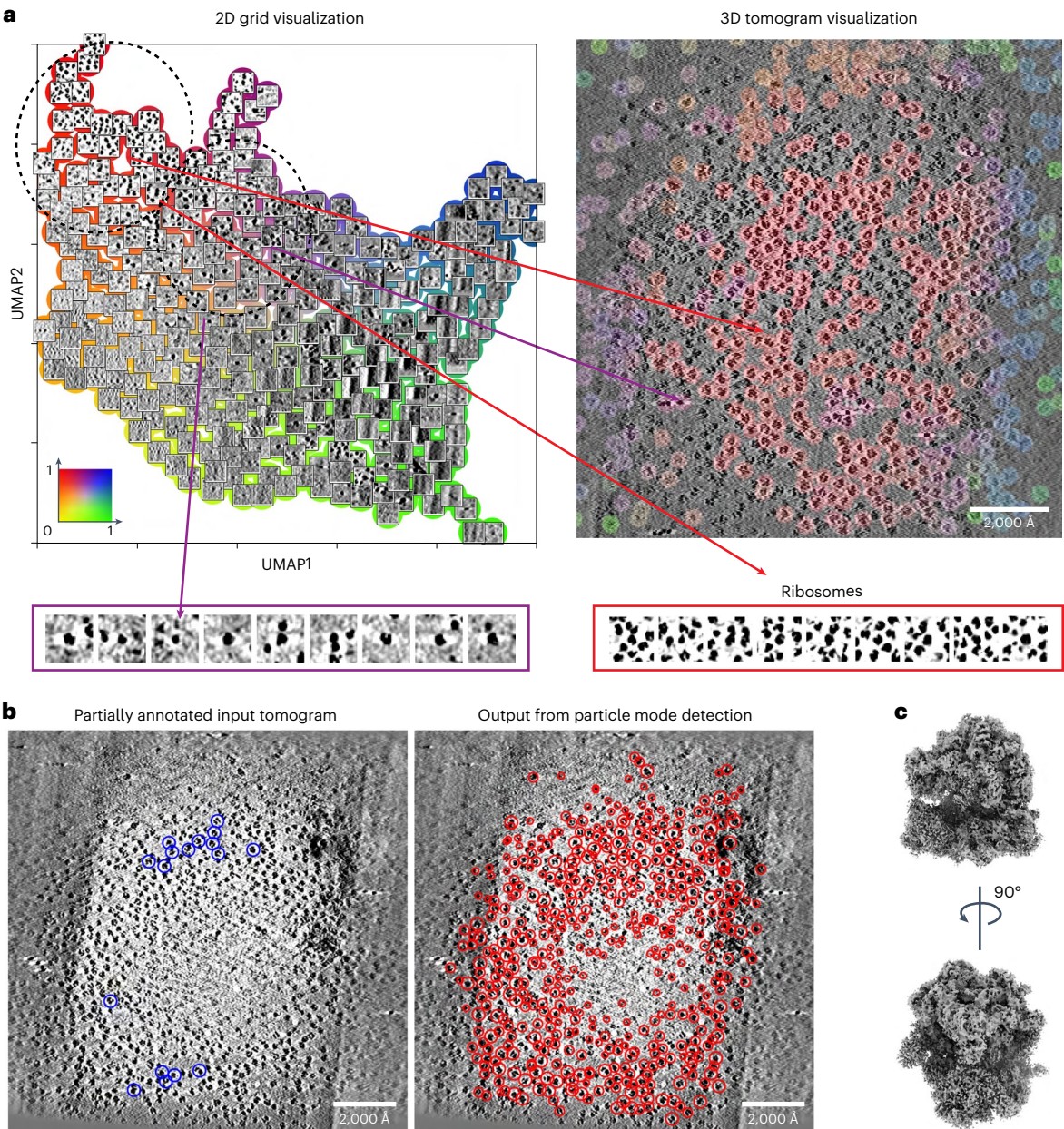

**Fig. 2 | Accurate detection of purified 70S ribosomes from densely populated tomograms.** We analyzed tilt series from the EMPIAR-10304 dataset using our content exploration and particle localization modules. **a**, 2D grid visualization of the embedding space (top, left) showing ribosomes (red), gold fiducials (purple), as well as other patches corresponding to contamination and background areas. A selection of patches containing gold fiducials (purple outline) and 70S ribosomes (red outline) is shown below. 3D tomogram visualization view (top, right) with voxels colored according to the embedding coordinates shown in the 2D grid visualization representation. The exploration module is able to successfully differentiate gold fiducials from ribosomes despite their similar size and shape. **b**, A partially annotated input tomogram showing 70S ribosomes (blue circles) used for training the refinement model (left). The trained model accurately identifies ribosomes (red circles) while ignoring gold beads and contamination/background areas (right). **c**, 5.0 Å resolution reconstruction of the 70S ribosome obtained from a total of 6,051 particles using SPT.

to 70S ribosomes (Methods), resulting in 195 positions that were used to train the detection module, which was then applied to all 65 tomograms in the dataset (Fig. 3b). A total of 23,285 particles were picked and subjected to 3D refinement and reconstruction in nextPYP, producing a 5.4 Å resolution reconstruction of the 70S ribosome (Fig. 3c). We also quantified the quality of particle picking by measuring F1 scores, evaluating changes in picking accuracy with respect to the number of annotations, and comparing detection performance against traditional template matching and crYOLO-3D (Extended Data Fig. 3b). To further assess MiLoPYP's performance on crowded samples, we analyzed tomograms from *Chlamydomonas reinhardtii* cells containing high concentrations

of the ribulose-1,5-bisphosphate carboxylase-oxygenase (RuBisCo) enzyme (available from EMPIAR-10694)[49]. 2D grid visualization of the learned embeddings from the exploration module revealed that the dataset is relatively homogeneous (Extended Data Fig. 5a). A total of 86 annotations were selected from the interactive session and used to train the particle localization module. After training, MiLoPYP identified a total of 36,345 particles, which were used to produce an 11.0 Å resolution structure of RuBisCo (Extended Data Fig. 5b). These results show MiLoPYP's ability to accurately detect targets in situ with minimal user intervention, addressing an important challenge in the CET/SPT structure determination pipeline.

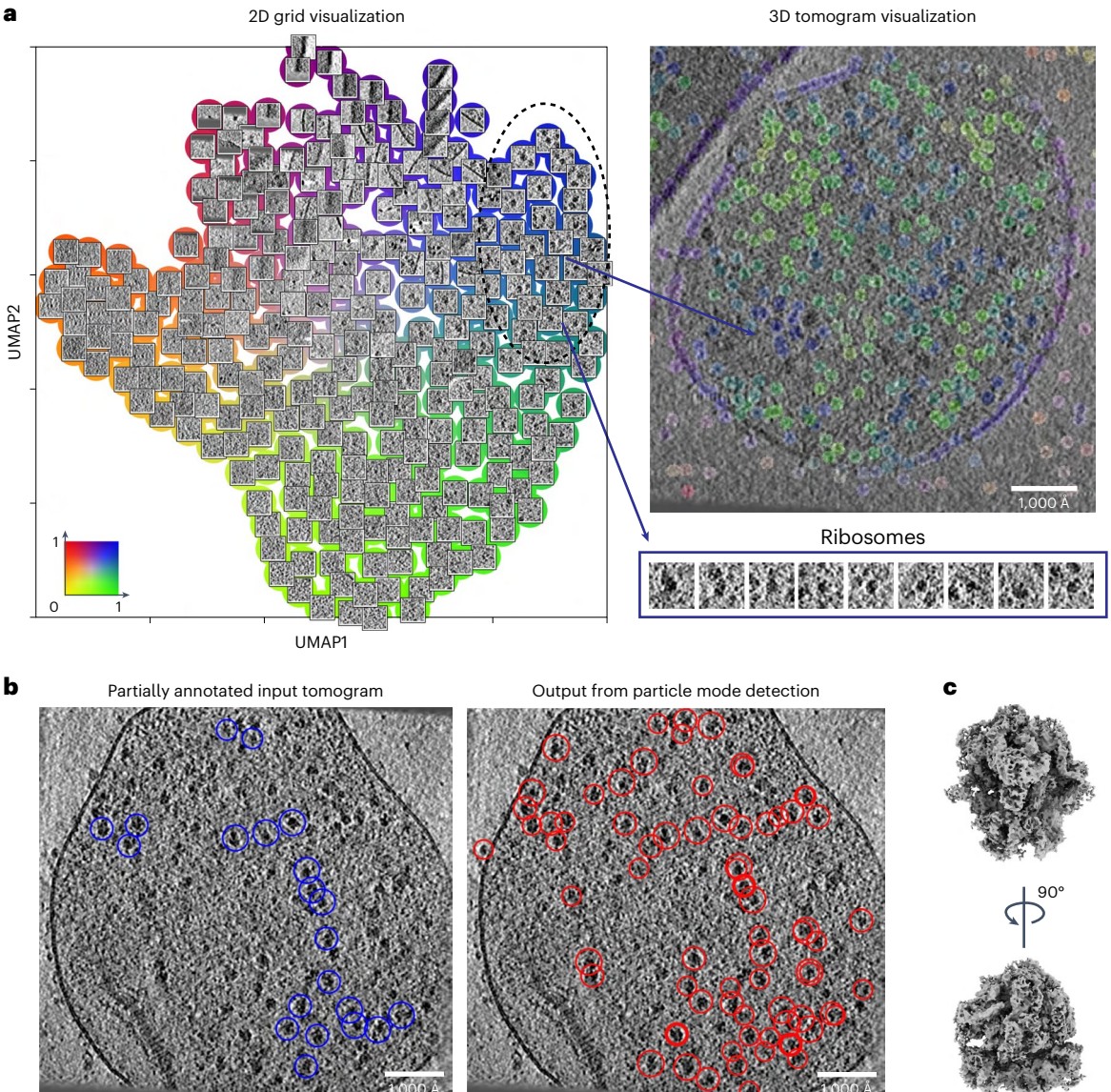

**Fig. 3 | Structural mapping and accurate identification of native ribosomes inside bacterial cells. a**, 2D grid visualization of embeddings obtained from tilt series downloaded from EMPIAR-10499 (top, left) showing ribosomes (blue), cellular membranes (purple/red) and other smaller-sized proteins (dark green). 3D tomogram visualization view (top, right) with voxels colored according to the embedding coordinates. A selection of patches containing 70S ribosomes is shown below (purple outline). **b**, A partially annotated input tomogram corresponding to 70S ribosomes (blue circles) used for training the refinement module (left). The trained model accurately identifies ribosomes (red circles) while ignoring membrane and background areas (right). **c**, 5.4 Å resolution SPT reconstruction of the 70S ribosome obtained from a total of 17,381 particles detected from 65 tomograms.

## Identification of native membrane proteins attached to viral particles

To test the ability of MiLoPYP to detect particles of more diverse shapes, we analyzed a dataset of severe acute respiratory syndrome coronavirus 2 (SARS-CoV-2) virus particles (EMPIAR-10453)[50]. 2D grid visualization of the learned embeddings showed two distinct populations of spikes present: membrane-attached spikes and free-floating spikes (Fig. 4a). To evaluate MiLoPYP's ability to focus on a specific structural pattern, we selected a total of 144 membrane-attached spikes from the interactive 3D session and used them to train the detection module. Running inference on the entire dataset yielded a total of 23,388 membrane-bound spikes (Fig. 4b). 3D refinement and classification in nextPYP revealed two conformations of membrane-attached spikes (open and closed) that were resolved to resolutions of 5.6 Å and 9.3 Å, from 9,194 and 3,740 particles, respectively (Fig. 4c). These results are consistent with findings reported in the original study where approximately

1,000 virions were manually selected followed by a labor-intensive geometry-based picking procedure[50]. These experiments demonstrate MiLoPYP's ability to accurately map and detect membrane-bound viral proteins in their native environment while requiring minimal user input.

## Simultaneous detection of multiple protein species from cellular lamellae

In addition to evaluating the discriminatory and localization capabilities of MiLoPYP on globular-shaped and native viral proteins, we further examined its ability to detect tubular-shaped complexes imaged by cryo-ET from focused ion beam/scanning electron microscopy (FIB/SEM)-milled samples. To do this, we analyzed a dataset of tomograms collected from thinned lamellae of mouse mammary gland epithelial EpH4 cells (EMPIAR-10987)[16]. As expected, the representation learned using the exploration module revealed the

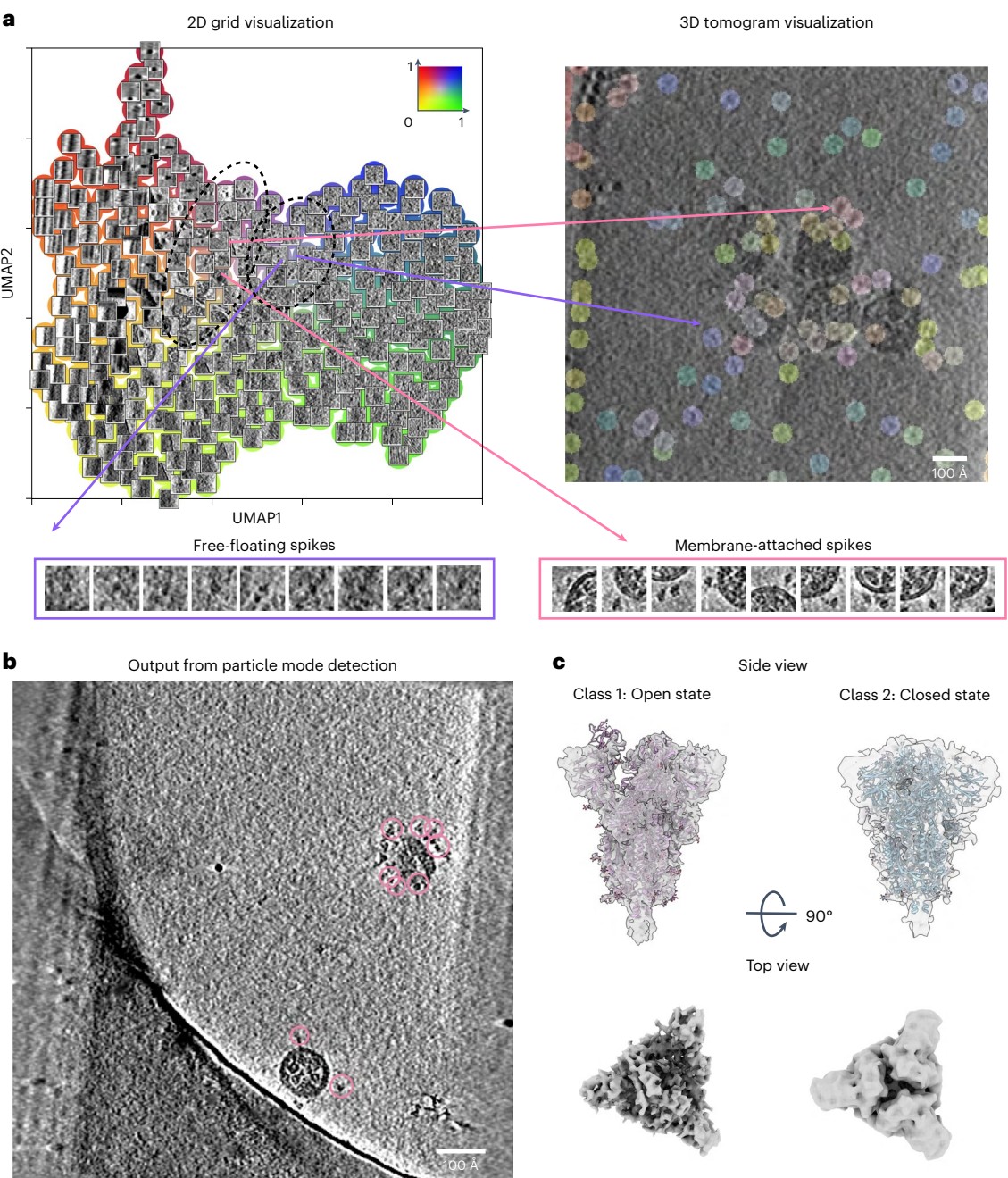

**Fig. 4 | Parsing of complex native environments and accurate identification of membrane-bound complexes.** We analyzed tilt series from EMPIAR-10453 containing SARS-CoV-2 viruses with spike protein attached to their surfaces. **a**, 2D grid visualization of SARS-CoV-2 sample showing patches corresponding to free-floating and membrane-bound spikes (top, left). 3D tomogram visualization with voxels colored according to the embedding coordinate values (top, right). Selections of patches containing free-floating spikes (purple outline) and membrane-attached spikes are shown below (pink outline). **b**, Membrane-attached SARS-CoV-2 spike localization after running the refinement module (pink circles). **c**, 5.6 Å and 9.3 Å resolution structures for the open and closed states determined by SPT from a total of 13,047 spikes extracted from 266 tomograms: open state reconstructed from 9,194 particles (left), and closed state from 3,740 particles (right). PDB coordinates 6VYB and 6VXX were fitted into the maps using rigid body fitting, respectively.

presence of membranes, fibers, microtubules and 80S ribosomes (Fig. 5a and Extended Data Fig. 6). We obtained coordinates for ribosomes and microtubules to train the refinement module by selecting corresponding subregions from the interactive 3D session, resulting in a total of 137 coordinates for ribosomes and 102 coordinates for microtubules (Fig. 5b). After training the refinement module using the ribosome positions, we obtained a total of 6,068 particles that resulted in a 8.6 Å resolution map of the 80S ribosome (Fig. 5c). We followed a similar procedure for the tubular-shaped particles, where coordinates for each microtubule were defined as the centers of their cross-sections, such that the number of points in a single microtubule was associated with its length (Extended Data Fig. 7). From 40 microtubules present across 21 tomograms, we successfully identified and traced 38 of them (Fig. 5b). Overlapping particles corresponding to microtubule segments were extracted and utilized for 3D refinement and reconstruction in nextPYP. From a total of 1,761 particles, we obtained a 37.0 Å resolution structure (Fig. 5d), demonstrating MiLoPYP's ability to detect protein species from a range of sizes and shapes from tomograms collected from FIB/SEM-milled lamellae samples.

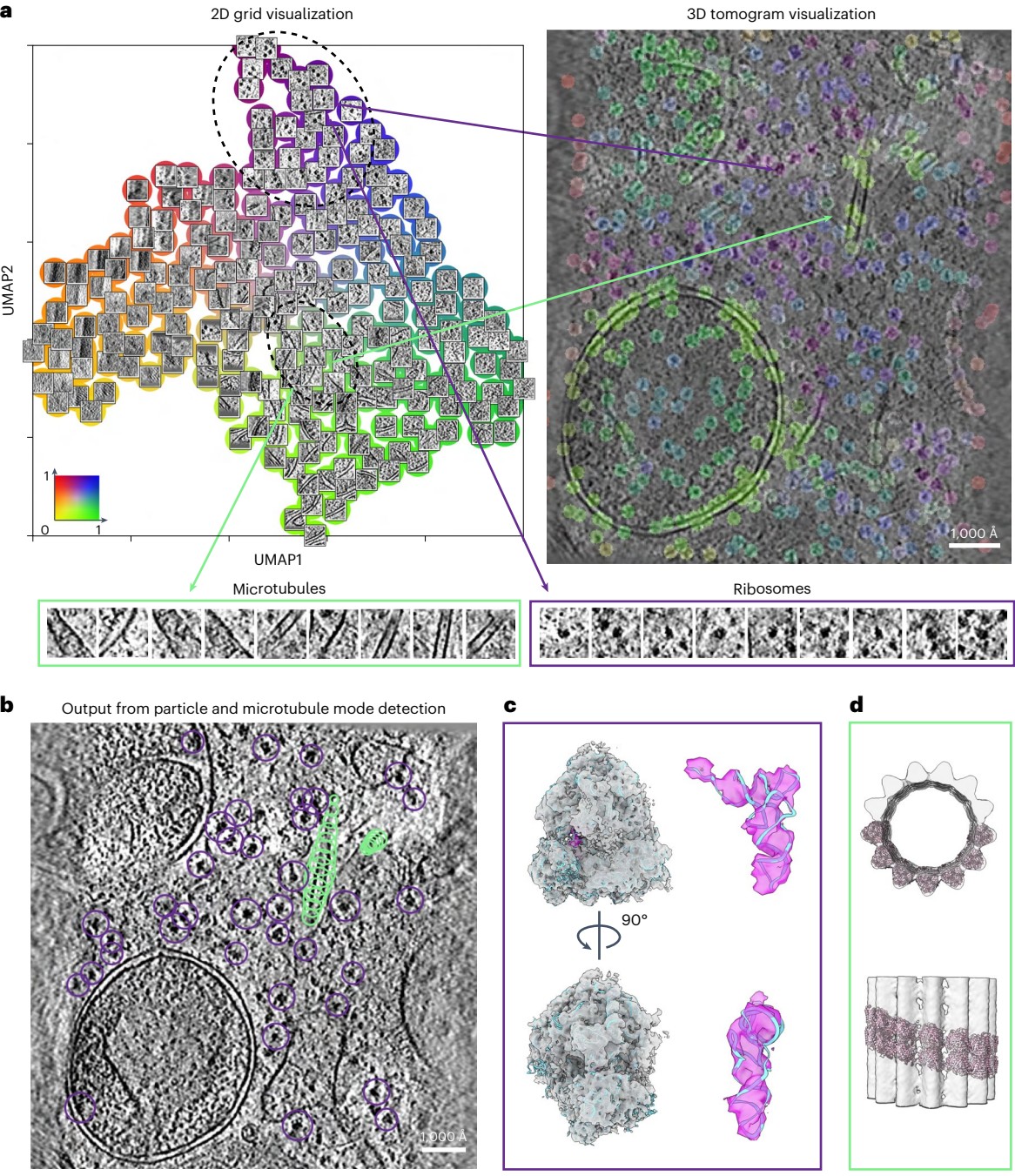

**Fig. 5 | Simultaneous identification and structure determination of multiple protein species from in situ lamellae. a**, 2D grid visualization of in situ tomograms from EMPIAR-10987 (top, left). The embedding space shows a variety of features and proteins including 80S ribosomes (purple) and microtubules (green). 3D tomogram visualization with voxels colored according to the embedding coordinates in the 2D grid visualization (top, right). A selection of patches containing microtubules (bottom, left) and 80S ribosomes (bottom,

right) are shown. **b**, After training and inference of the refinement module, MiLoPYP accurately localizes 80S ribosomes (purple) and microtubules (green). **c**, 8.6 Å resolution reconstruction of 80S ribosomes obtained from 4,570 particles. **d**, 37.0 Å resolution reconstruction of microtubules from 1,761 particles. PDB coordinates 6MTB and 6RZB were fitted into the maps using rigid body fitting, respectively.

## Accurate detection and localization of large membrane proteins in cells

We further examined MiLoPYP's ability to detect large membrane proteins from tomograms of FIB/SEM-milled *Saccharomyces cerevisiae* cells available from EMPIAR-11658. In addition to ribosomes, membranes and other cellular features, the representation learned using the exploration module revealed the presence of complexes bound to mitochondrial cristae consistent with the shape and dimensions

of ATP synthase (Fig. 6a). Corresponding coordinates selected using the interactive 3D session (Fig. 6b) were manually edited to ensure the quality of the training coordinates and used to train the refinement module. Compared to other datasets, this one required more particles to be selected (247) to achieve sufficient picking accuracy. A total of 4,105 particles were picked from 48 tomograms and subjected to 3D refinement and classification in nextPYP resulting in a 13.0 Å resolution map of ATP synthase from 2,577 particles (Fig. 6c), demonstrating

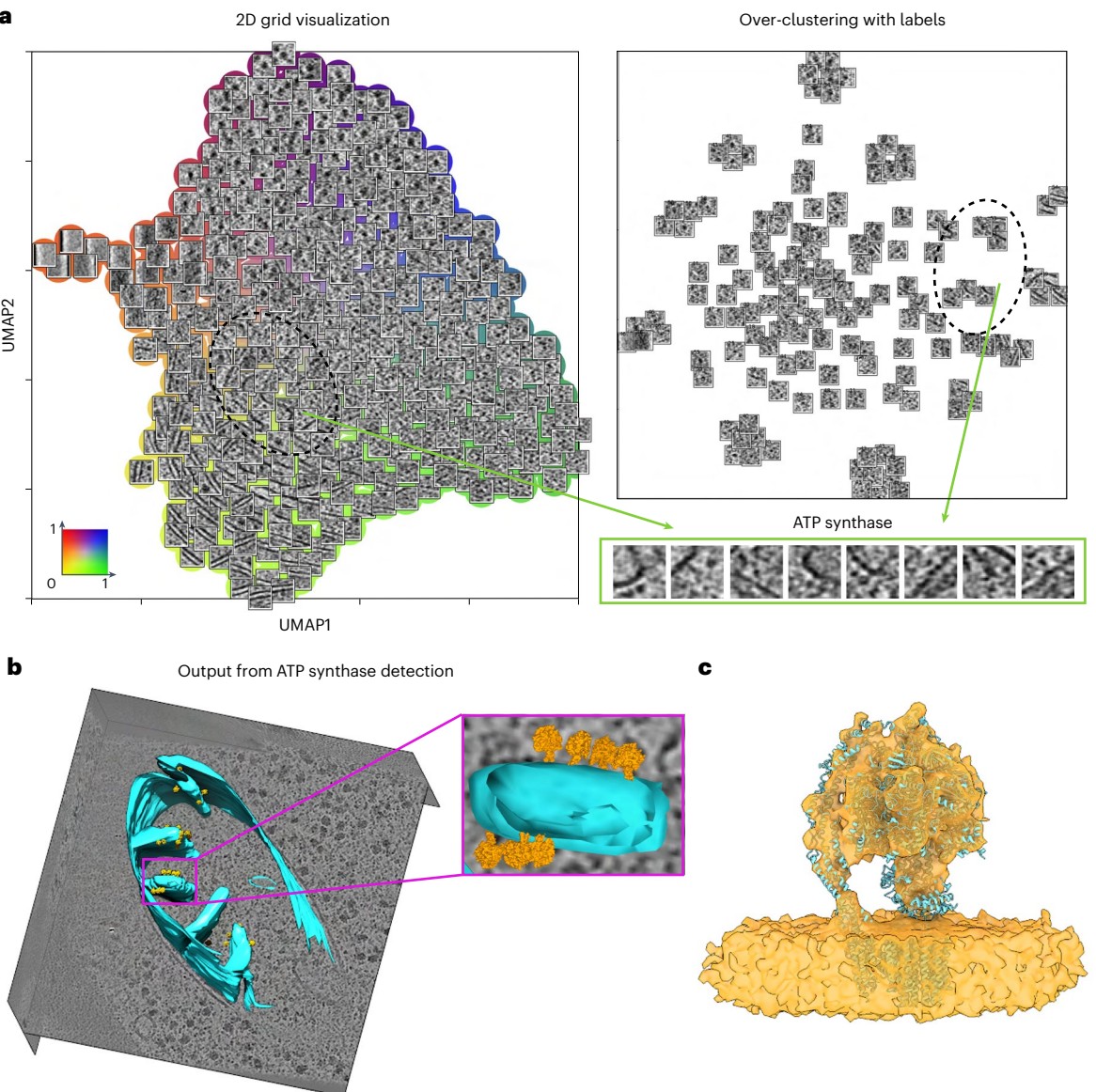

**Fig. 6 | Detection and structure determination of ATP synthase from in situ lamellae. a**, 2D grid visualization of tomograms from EMPIAR-11658 (left). The embedding space shows a variety of features and proteins including ATP synthase (green). Over-clustering with labels shows a group of patches corresponding to ATP synthase complexes (top, right). A selection of patches containing ATP synthase (bottom, left). **b**, Selected tomogram reconstruction with segmentation and the ATP synthase structure mapped back into the original positions. **c**, 13.0 Å resolution structure of ATP synthase obtained from 2,577 particles with PDB coordinates 8FL8 fitted into the map using rigid body fitting.

MiLoPYP's ability to detect large membrane proteins present at low concentration in cellular tomograms.

## Discussion

Recent advances in cellular sample preparation[10–12], tomographic data collection[13–17] and image processing[45,48] have established CET/SPT as the technique of choice for determining protein structures in their native state at high-resolution. One remaining technical challenge, however, has been the lack of computational tools to effectively sort through the intrinsic complexity of crowded cellular environments captured by CET. To address these challenges, we developed MiLoPYP, a robust and dataset-specific framework for molecular pattern mining that facilitates cellular content exploration and allows detection of a wide range of targets including globular-shaped, membrane-attached and fiber-like proteins. The framework is computationally efficient and requires minimal user intervention, making it suitable for processing large datasets that are required to determine high-resolution structures by SPT.

While MiLoPYP offers unique features and substantial advantages over existing approaches, it is also important to acknowledge some of its limitations. First, the representation learning module may struggle to distinguish proteins of similar shape or size, as exemplified in the detection of gold fiducials and ribosomes on EMPIAR-10304. Second, while self-supervised representation learning does not require a minimum number of exemplars to represent a single class, the 2D visualization step in MiLoPYP is prone to the class imbalance problem as dimensionality reduction techniques such as uniform manifold approximation (UMAP) and t-distributed stochastic neighbor embedding (t-SNE) are inevitably influenced by the number-of-neighbors parameter. Consequently, rare proteins present at lower concentrations may have their 2D representations blended with more abundant proteins of similar shape, preventing their correct identification. Provided that enough particles from each state or species are present, however, these problems can be dealt with during downstream SPT analysis using 3D classification techniques designed to characterize conformational variability.

Going forward, strategies for molecular pattern mining such as those implemented in MiLoPYP could be combined with emerging tools for 3D segmentation based on deep-learning that can accurately detect the position of subcellular features within tomograms such as membranes, filaments and organelles. Focusing the attention of MiLoPYP's exploration module on biologically relevant regions of the cell will result in improved pattern mining performance and higher detection accuracy, for example, allowing the routine localization of membrane proteins and other important targets attached to cytoskeleton components. Ultimately, the combination of deep-learning-based tools for image analysis with powerful strategies for SPT will facilitate the visualization of biomolecules in situ at high resolution.

In summary, MiLoPYP offers a convenient tool to map the interior of cells and find the location of multiple protein species within their native environment, as required for high-resolution analysis by SPT. Bypassing the need for laborious manual labeling, MiLoPYP can effectively map entire sets of tomograms, thus facilitating the interpretation, discovery and selection of target macromolecules. In addition to the precise identification of globular-shaped macromolecules, MiLoPYP can accurately detect membrane-bound and tubular-shaped complexes, converting it into a versatile tool for in situ molecular pattern mining. Importantly, the framework is computationally efficient and enables processing hundreds of tomograms, as needed for high-resolution SPT analysis. The code for MiLoPYP is distributed as open-source software, and the program has been incorporated into the user-friendly software package nextPYP.

## Online content

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

# Methods

## Self-supervised pattern mining and exploration module

The goal of the exploration module is to learn an embedding from the input set of tomograms and facilitate the identification of abundant protein species that could be used for high-resolution SPT analysis. This module is composed of three main steps: (1) 3D tomogram pre-processing, candidate coordinate generation, corresponding tilt series averaging and subtomogram slice extraction; (2) 2D CNN feature representation learning based on self-supervised contrastive learning; and (3) visualization of learned features.

**Preprocessing, coordinate generation and filtering.** A typical CET dataset, denoted as $\mathcal{D}$, consists of a collection of tomograms represented as $T_i \in R^{W \times H \times D}$, where $i$ ranges from 1 to $j$ ($j$ typically ranging from tens to a few hundred tomograms) and $W$, $H$ and $D$ are the tomogram dimensions. The preprocessing stage involves applying Gaussian denoising and histogram equalization techniques to enhance image contrast. 3D candidate coordinates are generated for each preprocessed tomogram $T_i$ using a DoG pyramid approach. To obtain these coordinates, the input tomogram $T_i$ is Gaussian filtered using progressively higher standard deviations $\sigma_i$ with $i = 1, \ldots, n$. This results in $n$ Gaussian smoothed tomograms, denoted as $\bar{T}_i$. A DoG representation is then obtained by calculating the difference between two Gaussian smoothed tomograms with different standard deviations: $DoG_i = \bar{T}_i - \bar{T}_{i+1}$. The final DoG representation is obtained by selecting the maximum voxel value across all levels of the pyramid followed by 3D NMS. Following this step, 3D coordinates are extracted based on voxel values that exceed a specified threshold determined automatically using 0.5 standard deviations above the mean voxel intensity. For each 3D coordinate $(x, y, z)$, its corresponding 2D coordinate $(x_\theta, y_\theta)$ in the tilted projection at angle $\theta$ is calculated as shown in equation (1):

$$(x_\theta, y_\theta) = \left( \left( x - \left\lfloor \frac{W}{2} \right\rfloor \right) \cos(\theta) + \left( z - \left\lfloor \frac{D}{2} \right\rfloor \right) \sin(\theta) + \left\lfloor \frac{W}{2} \right\rfloor, y \right). \quad (1)$$

Since tilted images within a −15° to +15° range contain higher contrast information, we calculate the projected 2D coordinates $(x_\theta, y_\theta)$ only for images in this angular range. Subsequently, patches centered at the calculated coordinates $(x_\theta, y_\theta)$ are extracted and averaged to obtain the final 2D tilt series input, denoted as $P_t$. In addition, we also extract $P_s$, consisting of the central tomographic slice at coordinates $(x, y, z)$. The pair $P_t$, $P_s$ serves as input to the neural network (Extended Data Fig. 1a). For datasets containing low-quality tilt series or substantial overlap in the $z$-direction due to crowding, only $P_s$ is used as input, rather than feeding the pair $P_t$, $P_s$.

NMS is an essential step aimed at selecting the most prominent locations based on nearby voxel values. This operation is performed in a greedy manner, starting from coordinates having the highest voxel values. To prevent the selection of neighboring voxels, a user-defined radius is used during NMS. Any voxel within this distance from an already selected coordinate is excluded from further consideration. The choice of radius is determined by the user and should be adjusted according to the expected particle density. For densely populated tomograms, a smaller radius is recommended to ensure accurate identification of individual particles and to minimize the risk of selecting nearby voxels.

## Overall architecture and contrastive learning loss function.

MiLoPYP uses 2D CNNs for feature representation learning. The overall architecture follows the Siamese neural network design, which contains two identical subnetworks with the same configuration and parameters[40,51]. During training, MiLoPYP takes as input two randomly augmented views $P_{t1}, P_{s1}$ and $P_{t2}, P_{s2}$ based on the original pair $P_t$, $P_s$. The two views are processed by an encoder network, $f$, consisting of a ResNet18-based backbone and a projection multilayer perceptron

(MLP) head, $h$. The encoder $f$ shares weights between the two views. A prediction MLP head $h$ is applied to the output of one view, and a stop-gradient operation is applied on the other side. The ResNet18 backbone consists of 18 layers[52], including one convolution layer with kernel size 7 × 7 and four residual blocks with kernel size 3 × 3, one average pooling layer and one fully connected layer. The projection MLP head is composed of three fully connected layers with one-dimensional (1D) batch normalization and rectified linear unit (ReLU) activation. The prediction MLP head is composed of two fully connected layers with 1D batch normalization and ReLU activation. The output vector from the encoder network of dimension 128 is denoted as $y_i \triangleq h(f(P_{t_i}, P_{s_i}))$ and the output vector from the prediction head $h$ of dimension 128 is denoted as $z_i \triangleq f(P_{t_i}, P_{s_i})$. CNNs are trained by maximizing the cosine similarity between the learned representations of positive pairs. Through this training process, MiLoPYP learns to embed macromolecules with shared structural similarities close to each other while placing dissimilar macromolecules farther apart in feature space.

Following ref. 40, we define the negative cosine similarity as in equation (2):

$$\mathcal{D}(y_1, z_2) = -\frac{y_1}{\| y_1 \|_2} \cdot \frac{z_2}{\| z_2 \|_2}, \quad (2)$$

where $\| \cdot \|_2$ represents the $L^2$ norm. The final symmetrized loss function has the form shown in equation (3):

$$\mathcal{L} = \frac{1}{2}\mathcal{D}(y_1, \text{stopgrad} z_2) + \frac{1}{2}\mathcal{D}(y_2, \text{stopgrad} z_1), \quad (3)$$

with a minimum possible value of −1. Here, 'stopgrad' refers to the stop-gradient operation where the model parameter-dependent variable $z$ is treated as a constant during back-propagation. By applying the stop gradient, the encoder function $f$ operating on the input pairs $P_{t1}, P_{s1}$ does not receive gradients from $z_2$ in the first term, but it does receive gradients from $y_1$ in the second term. The same principle applies to the input pairs $P_{t2}, P_{s2}$, where $f$ receives gradients from $y_2$ but not from $z_1$. This stop-gradient operation is crucial for preventing model degeneration when training models with only positive pairs.

## Model training, inference and data augmentation.

Training of the 2D CNN was performed for 300 epochs using stochastic gradient descent[53]. We initialize the network using pretrained ResNet18 weights on ImageNet from PyTorch. We use a batch size of 256 and a learning rate of 0.02. The learning rate has a cosine decay schedule. The weight decay is 0.0001 and the stochastic gradient descent momentum is 0.9. During training, a subset of the dataset $\mathcal{D}$ is used, which typically includes 5 to 10 tomograms. Training is performed on a single NVIDIA V100 GPU with 32 GB of RAM. Training for 300 epochs typically takes less than 2 h. After training, evaluation can be performed on the same subset of tomograms or on the entire dataset.

In self-supervised contrastive learning, the goal is to train the network in such a way that the representation of different augmented views of the same instance are maximally similar. To achieve this, careful selection of data augmentation techniques is essential to preserve the identity of each instance. In our approach, we apply several random augmentations during training. These include horizontal and vertical flipping, intensity jittering, random resize crop, corner dropout and rotations (multiples of 90°). Each of these augmentations serves a specific purpose in maintaining instance identities and promoting effective learning. Intensity jittering is used to vary the brightness and contrast of the input instances. Brightness is jittered from 0 to 0.5 and contrast is jittered from 0 to 0.2. By jittering these values within certain ranges, we enable the model to learn the shapes of instances irrespective of their gray value characteristics. Random resize crop is used to randomly crop a

portion of the input instances within a range of 0.7 to 1 times the original size, which is then resized back to the original size. This augmentation helps the model focus on learning the shapes of instances while minimizing sensitivity to their size variations. The choice of a larger lower bound (0.7) in the crop range allows the model to capture important shape information without completely disregarding size differences, which can be substantial in distinguishing macromolecules. Corner dropout involves randomly setting the voxel values to 0 within a subarea located at the corner of each input instance. This operation encourages the model to concentrate more on the central region (where particles are), enhancing the model's ability to capture important features. Rather than applying random rotations, we restrict rotation to multiples of 90°. This decision is based on experimental findings that random rotation, particularly with interpolation, can introduce distortions that may hinder shape learning. Using rotations that are multiples of 90° avoids interpolation artifacts and simplifies the resampling of voxel values.

**Dimensionality reduction, visualization and module output.** For 2D grid visualization, the 128-dimension learned representation is reduced to 2D using UMAP[54]. 2D embeddings were computed using version 0.5.0 of the Python implementation (https://github.com/lmcinnes/umap/) with $k = 40$ for the $k$-nearest-neighbors graph and a minimum distance parameter of 0.1. Using fewer neighbors may help reveal finer local structure, while the use of more neighbors will capture the overall structure with some loss of fine local structure (Extended Data Fig. 8). Grid visualization and 3D mapping tools are included with MiLoPYP, through the nextPYP and Phoenix (https://github.com/Arize-ai/phoenix/) applications.

To assign each learned embedding with a class label without knowing the actual number of classes $c$ in the dataset, we perform over-clustering and label reassignment. As the main objective is to explore protein variability within the dataset, instead of identifying the correct number of classes, grouping similar proteins into the same class is more important, which makes over-clustering a reasonable approach. For a set of embeddings, we first perform vanilla $k$-means using FAISS[55] with $k$ substantially larger than the actual potential number of clusters for the dataset, that is, $k \gg c$. Then, spectral clustering[56] is applied to $k$ cluster centers, and we further recluster these $k$ initial centers into $h$ classes, where $h$ is a user-defined value. Typically, $h$ is smaller than $k$ but still greater than the potential actual number of classes. Following the over-clustering and label reassignment step, the learned embeddings are classified into $h$ labels. Each class of embeddings is assigned a different color and users can select specific classes or individual patches to view in the 3D interactive session (Extended Data Figs. 4 and 6).

Through an interactive session, users can effectively identify targets of interest for further investigation. By selecting specific patches and their neighbors within a user-defined subregion or user-specified labels, the corresponding 3D coordinates can be obtained. It is important to note that as the subregions become bigger, the accuracy of selection diminishes due to the unsupervised nature of feature representation learning. To obtain a precise set of candidate coordinates, a refinement stage is necessary. The training process for the refinement model requires a small number of annotated examples. These examples can be derived by selecting candidate patches from the interactive session, eliminating the need for manual annotation.

## Few-shot particle localization module
Similarly to the exploration module, the particle localization module is composed of the following main steps (Fig. 1b and Extended Data Fig. 2): (1) 3D tomogram preprocessing with Gaussian denoising and voxel intensity rescaling using histogram equalization; (2) few-shot learning-based neural-network training; and (3) extraction of particle coordinates using NMS (for globular-shaped particles) or extraction of particle coordinates using polynomial fitting (for fiber-like proteins or microtubules; Extended Data Fig. 6). The particle localization module

offers the flexibility to be trained using partially annotated tomograms, which only need a limited number of labeled particles. These labels can be obtained either through the previous exploration module or by manual annotation. This approach enables efficient training and facilitates the accurate localization of particles with minimal requirements for labeled data.

**Framework for particle detection.** The objective of semi-supervised protein localization is to develop a model capable of detecting the locations of proteins in 3D tomograms, while learning from only a few annotated examples. Each tomogram $T_i$ contains a varying number of proteins, typically ranging from a few hundreds to a few thousands. In the semi-supervised protein identification scenario, the training set $\mathcal{D}_{\mathrm{tr}}$ consists of a single tomogram $T_{\mathrm{tr}}$ with a few annotated proteins, while the remaining tomograms serve as the testing set $\mathcal{D}_{\mathrm{te}}$. Unlike standard object detection algorithms that aim to localize objects of interest using bounding box locations and sizes, in cryogenic electron microscopy applications, the desired output for the particle detection task are the center coordinates of proteins. Hence, inspired by ref. 57, we train the particle detector using $\mathcal{D}_{\mathrm{tr}}$ to generate a center point heat map $\hat{Y} \in [0,1]^{C \times \frac{W}{R_1} \times \frac{H}{R_1} \times \frac{D}{R_2}}$, where $R_i$ represents the output stride and $C$ is the number of protein species. The output stride downsamples the prediction by a factor of $R_i$ on each dimension. In our case, we consider monodisperse samples, so we set $C = 1$ and we use $R_i = 1$. In subsequent sections, we omit the dimension for simplicity. We extend the approach used in ref. 58 to generate the ground-truth heat map $Y \in [-1,1]^{\frac{W}{R_1} \times \frac{H}{R_1} \times \frac{D}{R_2}}$ using the partially annotated tomogram. For each annotated center coordinate position $p = (x, y, z)$, we compute its downsampled equivalent as $\tilde{p} = (\lfloor \frac{x}{R_1} \rfloor, \lfloor \frac{y}{R_1} \rfloor, \lfloor \frac{z}{R_2} \rfloor)$. For each $\tilde{p}$ in $Y$, in the case of globular particle detection, we apply a Gaussian kernel $K_{xyz} = \exp(-\frac{(x-\tilde{p}_x)^2 + (y-\tilde{p}_y)^2 + (z-\tilde{p}_z)^2}{2\sigma_k^2})$, where $\sigma_k$ is determined by the particle size[58]. For the detection of tubular-shaped macromolecules, we set all voxels within a radius of 3 from $\tilde{p}$ to 1. The remaining unlabeled coordinates in $Y$ are assigned a value of −1.

**Feature extraction and center localization modules.** MiLoPYP's particle localization framework has three essential components (Extended Data Fig. 2): (1) an encoder–decoder feature extraction backbone, (2) a protein center localization module, and (3) a debiased voxel-level contrastive feature regularization module. Notably, although the network's input is a 3D tomogram, the feature extraction backbone primarily consists of 2D convolutional layers. Only the final two layers utilize 3D convolutional layers. This design is inspired by the manual particle picking process, where $xy$ slices contain the most useful information, while the $xz$ and $yz$ views provide limited information due to the missing wedge distortions. The extracted features are then directed to both the center localization module and the contrastive feature regularization module. The center localization module is trained using a novel objective function called positive unlabeled focal loss[59], which reduces overfitting when training with only positive data. On the other hand, the contrastive feature regularization module is trained using a debiased infoNCE loss[60], enabling it to learn the maximization of feature representation similarities for voxels belonging to the same protein class while minimizing the similarity between voxels corresponding to proteins versus non-protein regions. The combination of these two modules enables the accurate training of the protein localization model while minimizing annotations. Additionally, the contrastive regularization helps mitigate overfitting concerns commonly associated with positive unlabeled learning methods. By identifying particle centers on a voxel level, the trained model can precisely locate proteins even in crowded environments.

The feature extraction backbone of the model consists of a 2D UNet-based per-slice feature extraction backbone and a 3D convolutional layer for fusing the per-slice features into a single 3D representation. The number of encode blocks and decode blocks can be adjusted

by the user as hyper-parameters. For smaller-sized particles, it is recommended to use a maximum of four encode blocks. For larger-sized particles, five encode blocks are suggested. Each encode downsampling block includes two convolutional layers with a kernel size of $3 \times 3$, ReLU activation and max pooling with a kernel size of $2 \times 2$. Each decode upsampling block consists of one transpose convolutional layer with a kernel size of $2 \times 2$, two convolutional layers with a kernel size of $3 \times 3$, and ReLU activations. The 3D fusion layer is a convolutional layer with a kernel size of $3 \times 3 \times 3$, and ReLU activation. The overall architecture follows a Siamese network such that the input to the network is the tomogram $T$ and its augmented pair $\tilde{T}$. For the input tomogram $T \in \mathbb{R}^{W \times H \times D}$ and its augmented pair $\tilde{T}$, we denote the output from the feature extraction backbone as $M \in \mathbb{R}^{Ch \times \frac{W}{R} \times \frac{H}{R} \times \frac{D}{R}}$ and the augmented pair $\tilde{M}$. $M$ and $\tilde{M}$ are used to generate: (1) the output heat map $\hat{Y}$ and its augmented pair $\tilde{Y}$, and (2) the projected feature maps $F$ and $\tilde{F}$.

The center localization module is a 3D convolutional layer with kernel size $1 \times 1 \times 1$. Output of the center localization module is the heat map $\hat{Y}$. For the input tomogram $T$ and its output heat map $\hat{Y}$, protein localization can be viewed as a per-voxel classification problem such that each voxel $v_{i,j,k}$ at position $(i,j,k)$ is the input and the corresponding $\hat{y}_{i,j,k} \in [0,1]$ is the classification output. If we denote $p(v)$ as the underlying data distribution from which $v_{i,j,k}$ is sampled, $p(v)$ can be decomposed as in equation (4):

$$p(v) = \pi_p p_p(v|y=1) + \pi_n p_n(v|y=0), \qquad (4)$$

where $p_p(v|y=1)$ is the positive class conditional probability of protein voxels, $p_n(v|y=0)$ is the negative class conditional probability of background voxels, and $\pi_p$ and $\pi_n$ are the class prior probabilities. Subscripts $n$, $p$ and $u$ denote negative, positive and unlabeled, respectively. Denote $g : \mathbb{R}^d \to \mathbb{R}$, an arbitrary classifier that can be parameterized by a neural network, with $l(g(v) = \hat{y}, y)$ being the loss between model outputs $\hat{y}$ and ground truth $y$. When all the voxels are labeled, this is a binary classification problem that can be optimized using a standard positive-negative (PN) learning approach with the following risk minimization given by equation (5):

$$\tilde{R}_{pn} = \pi_p \tilde{R}_p^+(g) + \pi_n \tilde{R}_n^-(g), \qquad (5)$$

where $\tilde{R}_p^+(g)$ is the mean positive loss $\mathbb{E}_{v \sim p_p(v)}[l(g(v^p), y=1)]$ and can be estimated as $1/n_p \sum_{i=1}^{n_p} l(\hat{y}_p^i, 1)$, $\tilde{R}_n^-(g)$ is the mean negative loss $\mathbb{E}_{v \sim p_n(v)}[l(g(v^n), y=0)]$ and can be estimated as $1/n_n \sum_{i=1}^{n_n} l(\hat{y}_n^i, 0)$, and $n_p$ and $n_n$ are the number of positive and negative voxels. When only a few positive voxels are labeled and the remainder of the data are unlabeled, we reformulate the problem into the PU setting: the positively labeled voxels are sampled from $p_p(v|y=1)$, and the remaining unlabeled voxels are sampled from $p(v)$. As shown in ref. 61, by rearranging equations (4) and (5), we obtain $\pi_n p_n(v) = p(v) - \pi_p p_p(v)$ and $\pi_n \tilde{R}_n^-(g) = \tilde{R}_u^-(g) - \pi_p \tilde{R}_p^-(g)$. Therefore, we rewrite the risk minimization as in equation (6):

$$R_{pu} = \pi_p \tilde{R}_p^+(g) - \pi_p \tilde{R}_p^-(g) + \tilde{R}_u^-(g), \qquad (6)$$

with $\tilde{R}_u^-(g) = \mathbb{E}_{v \sim p(v)}[l(g(v), y=0)]$ and $\tilde{R}_p^-(g) = \mathbb{E}_{v \sim p_p(v)}[l(g(v), y=0)]$. To prevent overfitting in equation (6), we used non-negative risk estimation as in equation (7)[59]:

$$\tilde{R}_{pu} = \pi_p \tilde{R}_p^+(g) + \max\{0, \tilde{R}_u^-(g) - \pi_p \tilde{R}_p^-(g)\}. \qquad (7)$$

For globular-shaped particle detection, as the ground-truth heat map is splatted with Gaussian kernels, the labels are not strictly binary. Positive labels are split into two groups: true positives (tp) where $y_{i,j,k} = 1$, which is the center of each Gaussian kernel (protein center), and soft positives (sp) where $0 < y_{i,j,k} < 1$ (voxels that are close to the center). Unlabeled voxels are labeled as $-1$. With this, the positive distribution

$p_p(v)$ and positive-associated losses $\tilde{R}_p^+(g)$, $\tilde{R}_p^-(g)$ are decomposed into the following equation (8):

$$p_p(v) = \pi_{tp} p_{tp}(v|y=1) + \pi_{sp} p_{sp}(v|0 < y < 1),$$
$$\tilde{R}_p^+(g) = \pi_{tp} \tilde{R}_{tp}^+(g) + \pi_{sp} \tilde{R}_{sp}^+(g), \tilde{R}_p^-(g) = \pi_{tp} \tilde{R}_{tp}^-(g) + \pi_{sp} \tilde{R}_{sp}^-(g). \qquad (8)$$

We adopt voxel-wise logistic regression with focal loss for $l(g(v), y)$. Specifically, according to the following equation (9), we have:

$$\tilde{R}_{tp}^+(g) = (1 - \hat{y}_{ijk})^\alpha \log(\hat{y}_{ijk}), \tilde{R}_{sp}^+(g) = (1 - y_{ijk})^\beta (\hat{y}_{ijk})^\alpha \log(1 - \hat{y}_{ijk}),$$
$$\tilde{R}_{tp}^-(g) = \hat{y}_{ijk}^\alpha \log(1 - \hat{y}_{ijk}), \tilde{R}_{sp}^-(g) = (y_{ijk})^\beta (1 - \hat{y}_{ijk})^\alpha \log(\hat{y}_{ijk}), \qquad (9)$$
$$\tilde{R}_u^-(g) = (\hat{y}_{ijk})^\alpha \log(1 - \hat{y}_{ijk}).$$

By combining equations (7), (8) and (9), we obtain the final minimization objective as shown in equation (10):

$$\tilde{R}_{pu} = \pi_p(\pi_{tp} \tilde{R}_{tp}^+(g) + \pi_{sp} \tilde{R}_{sp}^+(g)) +$$
$$\max\{0, \tilde{R}_u^-(g) - \pi_p(\pi_{tp} \tilde{R}_{tp}^-(g) + \pi_{sp} \tilde{R}_{sp}^-(g))\}. \qquad (10)$$

For tubular-shaped macromolecule detection, there are only positive voxels and unlabeled voxels. Therefore, the only remaining terms are $\tilde{R}_{tp}^+(g)$, $\tilde{R}_{tp}^-(g)$, $\tilde{R}_u^-(g)$, and the final loss is equivalent to equation (7).

**Debiased contrastive regularization and final training loss.** As suggested in ref. 37, instead of using $M$, it is beneficial to map the representations to a new space through a projection head composed of a $1 \times 1 \times 1$ convolutional layer where a contrastive loss is applied. The contrastive regularization head is composed of a 3D convolutional layer with kernel size $1 \times 1 \times 1$. Output from the projection head is the projected feature map $F$ and $\tilde{F}$. Denote $f_{i,j,k} \in \mathbb{R}^{Ch}$ and $\tilde{f}_{i,j,k}$ as the feature vector at the $(i,j,k)$ position of the feature map $F$ and its augmented counterpart. There exists a total of $\frac{W}{R} \times \frac{H}{R} \times \frac{D}{R} = N$ such vectors. Each of these feature vectors is responsible for predicting $\hat{y}_{i,j,k}$. If $y_{i,j,k} = 1$, $m_{i,j,k}$ and its projection $f_{i,j,k}$ should encode particle-related features. For a partially annotated $T$, the voxel-level feature vector set $f$ can be separated into positive and unlabeled classes. Therefore, the voxel-level contrastive regularization loss is composed of positive supervised and unlabeled self-supervised debiased contrastive terms. For positive supervised debiased contrastive regularization, denote $\mathcal{F}^p = \{f_{i,j,k}^p : y_{i,j,k} = 1\}$ as the set of positive feature vectors obtained from $n_p$ annotated proteins and its augmented counterpart $\tilde{\mathcal{F}}^p = \{\tilde{f}_{i,j,k}^p : \tilde{y}_{i,j,k} = 1\}$, $\mathcal{F}^u = \{f_{i,j,k}^u : y_{i,j,k} < 1\}$ as the set of unlabeled (including the soft positives) feature vectors with a total of $n_n$. Because each tomogram contains up to a few hundred sub-volumes of the protein of interest, and each of these sub-volumes are the same protein with different relative orientations and are distorted in different ways, for a feature vector $f_i^p \in \mathcal{F}^p$, the remaining $2n_p - 1$ feature vectors $f_j^p, j = 1, \ldots, 2n_p - 1$ in $\mathcal{F}^p$ and $\tilde{\mathcal{F}}^p$ can be treated as naturally augmented pairs. Unlabeled feature vectors $f_k^u \in \mathcal{F}^u, k = 1, \ldots, 2n_n$, which include the augmented unlabeled features, are treated as negatives. However, because the unlabeled set $\mathcal{F}^u$ can contain positive feature vectors, the naive supervised contrastive loss as proposed in ref. 62 will be biased. Therefore, we adopt a modified debiased supervised contrastive loss based on equation (11)[63]:

$$\mathcal{L}_{sup}^{db} = \mathbb{E}\left[ -\log\left[ \frac{1/(2n_p - 1) \sum_{j=1}^{2n_p - 1} e^{f_i^{pT} f_j^p}}{1/(2n_p - 1) \sum_{j=1}^{2n_p - 1} e^{f_i^{pT} f_j^p} + g_{sup}(f_i^p, \{f_j^p\}_{j=1}^{2n_p - 1}, \{f_k^u\}_{k=1}^{2n_n})} \right] \right], \qquad (11)$$

where the second term in the denominator is given by equation (12):

$$g_{sup}(\cdot) = \max\left\{ \frac{1}{\pi_n}\left( \frac{1}{2n_n} \sum_{k=1}^{2n_n} e^{f_i^{pT} f_k^u} - \pi_p \frac{1}{2n_p - 1} \sum_{j=1}^{2n_p - 1} e^{f_i^{pT} f_j^p} \right), e^{-1/t} \right\}, \qquad (12)$$

with $\pi_n$ and $\pi_p$ being the class prior probabilities and $t$ the temperature.

Regarding the unlabeled self-supervised debiased contrastive regularization, for the unlabeled feature vector $f_k^u$, the only known positive is its augmented pair $\tilde{f}_k^u$, and the remaining vectors are treated as negatives. Denote $\{f_l^r\}_{l=1}^{2N-2}$ as the set of remaining vectors. The resulting contrastive loss for an unlabeled feature vector is given by equation (13):

$$\mathcal{L}_{\text{unsup}} = \mathbb{E}\left[ -\log \left[ \frac{e^{f_k^{uT}\tilde{f}_k^u}}{e^{f_k^{uT}\tilde{f}_k^u} + g_{\text{unsup}}(f_k^u, \tilde{f}_k^u, \{f_l^r\}_{l=1}^{2N-2})} \right] \right]. \qquad (13)$$

Similarly to equation (12), $g_{\text{unsup}}(\cdot)$ involves class prior probabilities. However, the actual class of the unlabeled feature vectors is unknown. Therefore, we divide the unlabeled voxels into three groups: pseudo-positive, pseudo-negative and neutral. Pseudo-positives are voxels with predicted particle probability greater than 0.9. Pseudo-negatives are voxels with predicted particle probability smaller than 0.3. The remaining voxels are neutral. For the pseudo-positives loss $\mathcal{L}_{\text{unsup}}^p$, $g_{\text{unsup}}^p$ in the denominator is expressed as shown in equation (14):

$$g_{\text{unsup}}^p(\cdot) = \max\left\{ \frac{1}{\pi_n}\left( \frac{1}{2N-2}\sum_{l=1}^{2N-2} e^{f_k^{uT}f_l} - \pi_n e^{f_k^{uT}\tilde{f}_k^u} \right), e^{-1/t} \right\}, \qquad (14)$$

which uses the negative class prior probability $\pi_n$. For the pseudo-negatives loss $\mathcal{L}_{\text{unsup}}^n$, $g_{\text{unsup}}^n$ in the denominator is expressed as shown in equation (15):

$$g_{\text{unsup}}^n(\cdot) = \max\left\{ \frac{1}{\pi_p}\left( \frac{1}{2N-2}\sum_{l=1}^{2N-2} e^{f_k^{uT}f_l} - \pi_p e^{f_k^{uT}\tilde{f}_k^u} \right), e^{-1/t} \right\}, \qquad (15)$$

which uses the positive class prior probability $\pi_p$. The loss for neutrals is calculated as the weighted average based on the probabilities of the feature vector belonging to the positive class as given by equation (16):

$$\mathcal{L}_{\text{unsup}}^{db} = \hat{Y}\mathcal{L}_{\text{unsup}}^p + (1 - \hat{Y})\mathcal{L}_{\text{unsup}}^n. \qquad (16)$$

We added a consistency regularization loss using the mean-squared error (MSE) between the output heat map $\hat{Y}$ and its augmented version $\tilde{Y}$ such that the probability of a voxel containing a protein should be invariant to augmentations as shown in equation (17):

$$\mathcal{L}_{\text{cons}} = \text{MSE}(\hat{Y}, \tilde{Y}). \qquad (17)$$

The final training objective is shown in equation (18):

$$\mathcal{L} = \tilde{R}_{pu} + \lambda_1(\mathcal{L}_{\text{sup}}^{db} + \lambda_2\mathcal{L}_{\text{unsup}}^{db}) + \lambda_3\mathcal{L}_{\text{cons}}, \qquad (18)$$

where $\lambda_1$ is the weight of the total contrastive module, $\lambda_2$ is the weight of the unsupervised contrastive loss, and $\lambda_3$ is the weight of consistency regularization. The resulting loss serves two purposes: (1) the contrastive term maximizes similarities for encoded features belonging to the same group (particle and background) and minimizes such similarities if features are from different groups; and (2) the heat map loss term forces predicted particle probabilities to be higher when they are closer to the true center location. To remove duplicate predictions, NMS is applied to the predicted heat map using 3D maximum pooling with a user-defined kernel size. For tubular-shaped macromolecules, we fixed the kernel size to 3 as we need to extract more coordinates for the polynomial fitting step.

**Post-processing for tubular-shaped macromolecules.** We apply a post-processing step specifically designed to detect tubular-shaped macromolecules. The workflow consists of three main stages: (1) grouping of extracted coordinates, (2) second-order polynomial fitting, and (3) resampling based on the fitted polynomial. In the grouping stage,

we construct an undirected graph using the extracted coordinates, where each vertex represents a coordinate and an edge exists between two vertices if their Euclidean distance falls within a user-defined cutoff distance. Connected components in the graph correspond to groups of coordinates. For each connected component, we perform second-order polynomial fitting separately for the $xy$ and $yz$ dimensions as given by equation (19):

$$\begin{aligned} \hat{x} &= a_0 + b_0 y + c_0 y^2, \\ \hat{z} &= a_1 + b_1 y + c_1 y^2. \end{aligned} \qquad (19)$$

The quality of the fit is assessed by measuring the residual between the fitted points and the actual values. Additionally, we calculate the maximum curvature of the fitted polynomial using equation (20):

$$k = \max\left( 2a / ((1 + 2ay + b)^2)^{\frac{2}{3}} \right). \qquad (20)$$

If the fitted residual and maximum curvature are both below user-defined cutoff values, we consider the set of coordinates as potential candidates for being tubular-shaped macromolecules. Finally, we resample the coordinates based on the fitted polynomial. This involves determining the minimum and maximum values of $y$ among the extracted coordinates and performing uniform sampling of $\hat{y}$ values with a step size of 2. The corresponding $\hat{x}$ and $\hat{z}$ values are then calculated using the fitted polynomial in equation (19). The resulting resampled 3D coordinates $(\hat{x}, \hat{y}, \hat{z})$ are used as the final coordinates identifying the position of tubular-shaped macromolecules.

**Comparison with state-of-the-art approaches.** For comparing the performance of MiLoPYP against template matching and crYOLO-3D, we used tomograms from EMPIAR-10304 and EMPIAR-10499. For EMPIAR-10304, we used EMAN2's implementation of template matching (e2spt_tempmatch.py)[18]. For EMPIAR-10499, we manually picked particles and used them to generate an initial reference (pixel size = 10 Å, box size = 34). This reference was low-pass filtered to 40.0 Å and used for template matching as implemented in Warp[64]. Particles were selected at positions that had _rlnAutopickFigureOfMerit values greater than 13. For crYOLO-3D, we retrained the model on each dataset using manually picked particles across ten consecutive slices (selecting 20–30% of particles) and using tomograms with different defocus values. Our results show what MiLoPYP has higher accuracy (F1 scores) when compared to template matching and crYOLO-3D (Extended Data Fig. 3).

We also compared the performance of MiLoPYP against three other deep-learning-based approaches: DeepFinder[31], DeePiCt[32] and TomoTwin[34]. To ensure a thorough and meaningful comparison, we used tomograms from four different datasets representing synthetic datasets (SHREC-21) and real datasets (EMPIAR-10304, EMPIAR-10453 and EMPIAR-10987; Supplementary Table 1 and Extended Data Figs. 9 and 10). When running TomoTwin, we used the latest pretrained model weights and followed the instructions from the reference-based particle picking tutorial. For DeepFinder, F1 numbers for the SHREC-21 dataset were obtained from the SHREC-21 challenge results; for the EMPIAR datasets, we used the pretrained model weights (as the model was trained to detect four classes, we only report F1 numbers corresponding to the ribosome class). For DeePiCt, we used the pretrained model to detect ribosomes from EMPIAR-10304 and EMPIAR-10987. As DeePiCt was only trained to detect ribosomes, membranes, microtubules and fatty acid synthase, we could not apply this method to the SHREC-21 dataset (F1 scores reported as 'not applicable'). Also, because neither DeepFinder nor DeePiCt was trained to detect native viral spikes, we did not obtain meaningful results when analyzing tomograms from EMPIAR-10453 using these methods (F1 scores reported as 'not applicable'). As expected, our experiments show that on the synthetic

SHREC-21 dataset the fully supervised method DeepFinder performs best on average (as more data are available for training compared to semi-supervised methods), while MiLoPYP and TomoTwin outperform DeepFinder and DeePiCt on the three EMPIAR datasets, with MiLoPYP outperforming TomoTwin by 7% (F1 scores).

**Model training, inference, evaluation and ablation studies.** During training, instead of using the whole tomogram, we cropped subtomograms of size 64 × 64 × 5 as input to the network in batches of 2. Training time is thus independent of the input size. Inference is performed on the entire tomogram. The proposed framework is trained in an end-to-end manner using Adam optimizer with default parameter values and an initial learning rate of 0.001. We decrease the learning rate by a factor of 10 every 5 epochs. Training takes around 3 to 5 minutes for 10 epochs, and inference on each full tomogram takes less than a second on an NVIDIA Tesla V100 GPU. We used experimentally determined values: $\lambda_1 = 0.1$, $\lambda_2 = 0.5$ and $\lambda_3 = 0.1$.

For datasets where manual labels were available, we evaluated the quality of detection in terms of precision and recall scores. We followed the same definition of these metrics as in ref. 27. To account for small variations in the detected particle centers, instead of looking at a single pixel, we also look at pixels located within a certain radius from the center. If the detected particle position is within a certain radius of a ground-truth particle position, we considered it as a true positive match. Let $TP(k)$ be the number of true positives in the top $k$ predictions. Precision is defined as the fraction of predictions that are correct and recall is defined as the fraction of labeled particles that are retrieved in the top $k$ predictions. With this, precision and recall are calculated as shown in equation (21):

$$TP(k) = \sum_i^k y_i,$$
$$Pr(k) = \frac{TP(k)}{k}, \qquad (21)$$
$$Re(k) = \frac{TP(k)}{\sum_i^n y_i}.$$

To assess the effectiveness of the proposed contrastive learning regularization and positive unlabeled formulation, we conducted ablation studies on the voxel-level contrastive learning module and the positive unlabeled learning component, using the EMPIAR-10304 (Extended Data Fig. 3a) and EMPIAR-10499 (Extended Data Fig. 3b) datasets. In the first ablation study, we removed the voxel-level contrastive module and the corresponding term in the loss function. As a result, we observed a degradation in performance for both datasets, with a more substantial drop in performance observed in the more complex in situ dataset EMPIAR-10499. This demonstrates that inclusion of contrastive regularization enhances the feature learning process and facilitates the detection of particles even when limited training samples are available. For the second ablation study, we removed the positive unlabeled module by treating all unlabeled regions as negatives and utilized a standard focal loss while maintaining the contrastive module unchanged. Similarly, we observed a substantial deterioration in performance (Extended Data Fig. 3). This highlights the importance of debiasing in scenarios where annotated data are scarce. Together, these experiments establish the substantial contributions of the voxel-level contrastive learning and the positive unlabeled learning components, emphasizing their effectiveness in enhancing feature learning, mitigating biases and enabling robust particle detection, particularly when confronted with limited annotated data.

**Contribution of localization module to detection accuracy.** To assess the performance improvement enabled by the protein-specific particle localization module, we compared the precision, recall and F1 scores of particles detected using the exploration module alone and particles detected after training of the localization module. Results on tilt series and tomograms from EMPIAR-10304, EMPIAR-10499, EMPIAR-10453 and EMPIAR-10987 show substantial gains in all accuracy metrics when adding the localization module (Supplementary Table 2).

**Dataset description and SPT details**
Tilt series from EMPIAR-10304 were downloaded from the EMPIAR database and used to reconstruct tomograms at a pixel size of 25.2 Å in nextPYP[45]. These tomograms were used as input to MiLoPYP's exploration and refinement modules. Evaluation of the refinement module on 11 tomograms produced a total of 8,965 particles that were subjected to SPT analysis in nextPYP. Particles were extracted using a binning factor of 4 and subjected to reference-based alignment using a map downloaded from EMD-23358 and a maximum resolution for refinement of 22 Å, followed by particle filtering according to the particle score distribution. A subset of 6,051 good-quality particles was kept for further processing. Particles were re-extracted without binning, and constrained particle and micrograph alignment using a shape mask was performed until no further improvements in resolution were observed. After per-particle contrast transfer function (CTF) refinement and an additional round of constrained particle and micrograph refinement, we obtained a final reconstruction of the 70S ribosome at 5.0 Å resolution (Fig. 2c and Supplementary Fig. 1a).

For EMPIAR-11694, a total of 36,345 particles corresponding to RuBisCo enzymes were identified using a pixel size of 13.68 Å. Reference-based refinement in nextPYP gave an 11.0 Å resolution map from 35,352 particles (Extended Data Fig. 5b and Supplementary Fig. 1b). D4 symmetry was applied to the final reconstruction.

For EMPIAR-10499, a total of 23,285 particles were identified from 65 tomograms reconstructed using a pixel size of 13.6 Å. A similar procedure to the one used for EMPIAR-10304 was conducted in nextPYP using a map downloaded from EMD-11650 as a reference, resulting in a 5.4 Å resolution reconstruction of the 70S ribosome from 17,381 particles (Fig. 3c and Supplementary Fig. 1c).

For EMPIAR-10453, a total of 23,388 particles were identified from 266 tomograms reconstructed using a pixel size of 15.95 Å. All particles were used for reference-based refinement in nextPYP using a map downloaded from EMD-11347 and a maximum resolution during refinement of 20 Å. Constrained 3D classification without alignment was performed to remove particles containing projections that were either too close to each other or too close to gold beads. A total of 13,047 particles were kept and further refined using constrained refinement with a shape mask. 3D classification without alignment using three classes and a focus mask covering the receptor-binding domain (RBD) was performed. Class 1 only had a few particles resembling gold beads and was discarded. Class 2 had 9,194 particles showing an open spike conformation with one RBD pointed upwards, and class 3 containing 3,740 particles showed a closed spike with the three RBDs pointing down. Further constrained refinement was performed on these two classes yielding a final reconstruction for the open state of the SARS-CoV-2 spike of 5.6 Å resolution (no symmetry was applied) and 9.3 Å resolution for the closed state (C3 symmetry was applied; Fig. 4c and Supplementary Fig. 1d,e).

For EMPIAR-10987, a total of 6,068 particles corresponding to 80S ribosomes were identified from 21 tomograms reconstructed using a pixel size of 15.95 Å. A similar processing procedure as that used for EMPIAR-10304 was followed in nextPYP using a map downloaded from EMD-33118 as a reference, yielding a 8.6 Å resolution map from 4,570 particles (Fig. 5c and Supplementary Fig. 1f). From the same set of 21 tomograms, a total of 1,761 particles corresponding to microtubules were identified and subjected to 3D refinement in nextPYP. All particles were used for reference-based refinement using a map downloaded from EMD-10061 as a reference, resulting in a 37.0 Å map (Fig. 5d and Supplementary Fig. 1g). No constrained 3D classification was conducted because of the limited number of particles. Helical symmetry was applied to the final map.

For EMPIAR-11658, a total of 4,105 particles corresponding to ATP synthase were identified from 48 tomograms reconstructed using a pixel size of 15.68 Å that contained mitochondrial membranes. A similar processing procedure as that used for EMPIAR-10987 was followed in nextPYP using a map downloaded from EMD-28809 as a reference, yielding a 13.0 Å resolution map from 2,577 particles (Fig. 6c and Supplementary Fig. 1h).

In all cases, map resolution was measured using the 0.143 cutoff of Fourier shell correlation curves between half-maps. Additional data processing statistics and timing information are included in Supplementary Table 3.

## Reporting summary

Further information on research design is available in the Nature Portfolio Reporting Summary linked to this article.

## Data availability

This study utilized raw tilt series data available from the EMPIAR database under accession numbers EMPIAR-10304, EMPIAR-10453, EMPIAR-10499, EMPIAR-10694, EMPIAR-10987 and EMPIAR-11658. Cryogenic electron microscopy density maps produced in this study were deposited in the Electron Microscopy Data Bank under accession numbers EMD-45261 for EMPIAR-10304, EMD-45266 for EMPIAR-10499, EMD-45267 and EMD-45268 for EMPIAR-10453, EMD-45269 for EMPIAR-10694, EMD-45270 and EMD-45271 for EMPIAR-10987 and EMD-45272 for EMPIAR-11658.

## Code availability

The open-source code for MiLoPYP is available at https://github.com/nextpyp/cet_pick/. The documentation and a user guide are available at https://nextpyp.app/milopyp/.

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

## Acknowledgements

Research reported in this publication was supported by the National Institute of General Medical Sciences of the National Institutes of Health (award no. R01GM141223 to A.B.). The content is solely the responsibility of the authors and does not necessarily represent the views of the National Institutes of Health. This study has been made possible in part by CZI grant DAF2021-234602 and a grant from the Chan Zuckerberg Initiative DAF (https://doi.org/10.37921/181830egjglp) and an advised fund of Silicon Valley Community Foundation (https://doi.org/10.13039/100014989). This study was partially supported by the National Institute of Allergy and Infectious Diseases of the National Institutes of Health (award no. U54-AI170752 to A.B.). This study utilized the computational resources offered by Duke Research Computing (http://rc.duke.edu/). We thank T. Futhey, C. Kneifel, K. Kilroy, M. Newton, V. Orlikowski, T. Milledge, J. Dorff, Z. Hill and D. Lane from the Duke Office of Information Technology and Research Computing for providing assistance with the computing environment.

## Author contributions

Q.H., Y.Z. and A.B. developed and implemented MiLoPYP. Q.H. and Y.Z. performed data processing. Q.H., Y.Z. and A.B. contributed to writing the paper and prepared the figures. Q.H. and A.B. conceived the project.

## Competing interests

The authors declare no competing interests.

## Additional information

**Extended data** is available for this paper at https://doi.org/10.1038/s41592-024-02403-6.

**Correspondence and requests for materials** should be addressed to Alberto Bartesaghi.

**a**

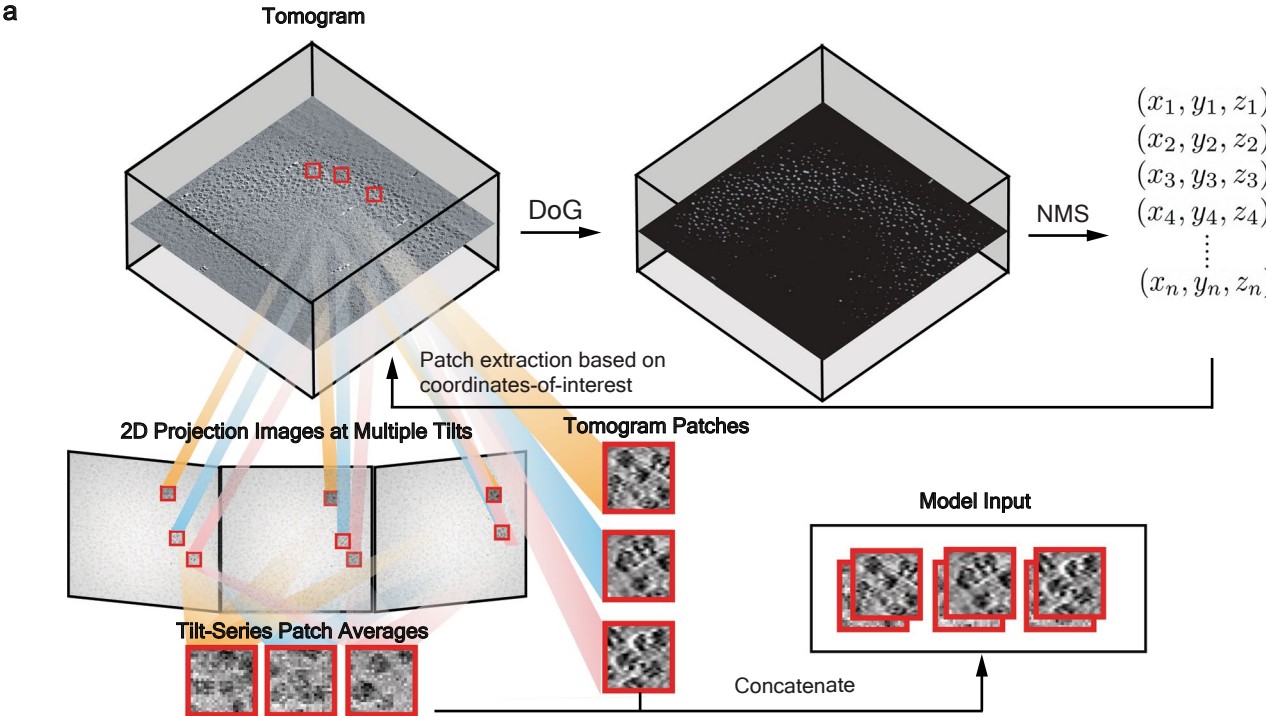

**b**

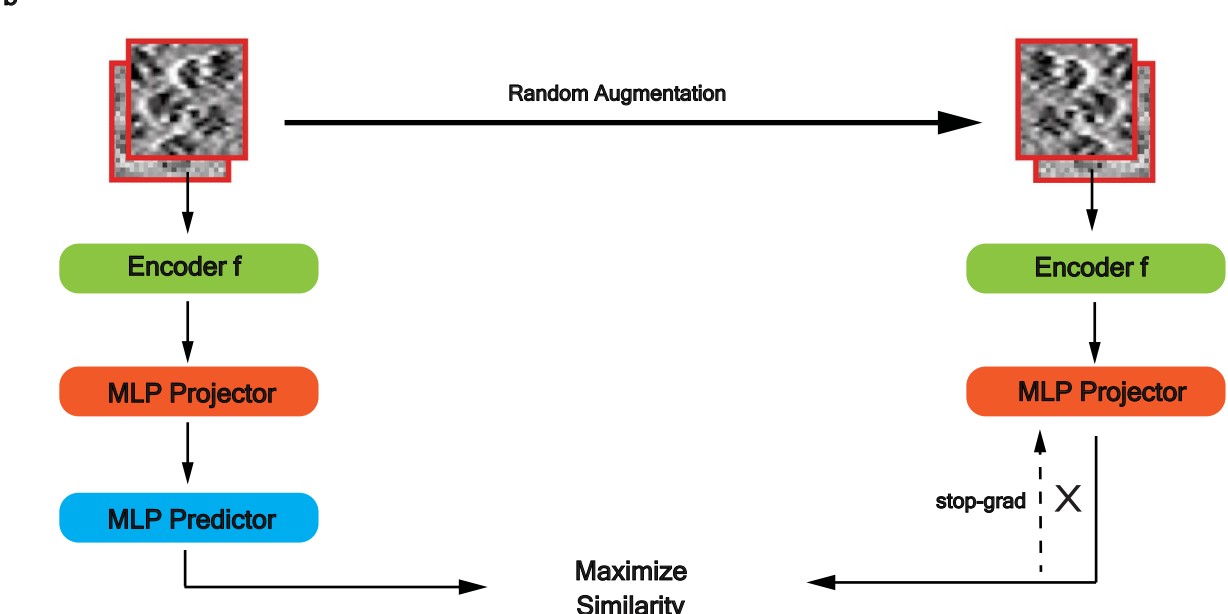

**Extended Data Fig. 1 | Overall architecture of self-supervised cellular content exploration module. a**) Preprocessing step which involves initial coordinate calculation based on Difference of Gaussians (DoG) and patch extraction from corresponding locations in 3D tomogram and 2D tilt series. 2D patches obtained by averaging tilt series projections centered on each position in 3D and corresponding sub-tomogram projections in the Z-direction are concatenated and serve as input to train the model. A non-maximum suppression (NMS) module is used to identify the final candidate positions. **b**) MiLoPYP's contrastive representation learning network follows a simple Siamese 'SimSiam' architecture. The input and its augmented version are used as inputs and the network learns to maximize the similarity between the pairs. The network is composed of an ResNet-based encoder, a multi-layer perceptron (MLP) projector and a MLP predictor. Stop-gradient is used to avoid model collapse.

a

b

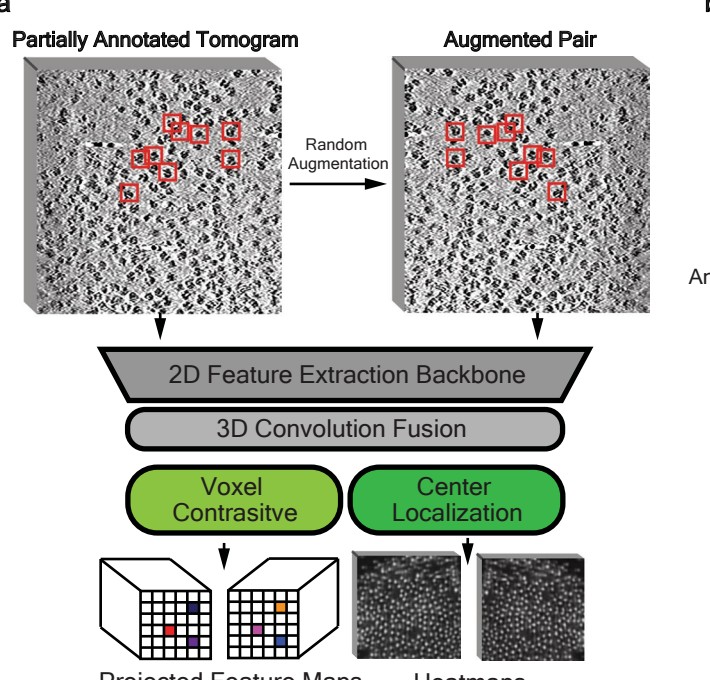

**Extended Data Fig. 2 | Overall architecture of semi-supervised particle localization module. a)** Partially annotated tomograms and their augmented versions are used as input to a network that uses a combination of 2D and 3D convolutional layers. The 2D CNN-based feature extractor follows a standard encoder-decoder architecture. The feature extractor is applied to each slice of the tomogram. The extracted features are then fused together through 3D convolutional layers. The fused 3D features are used for center coordinate heatmap prediction and voxel-level contrastive learning. During inference, only the heatmaps are used for particle identification. **b)** Schematic of contrastive learning module. Stars represent 3D particles and their naturally augmented positive pairs (particles belonging to the same class). Feature vectors at the star locations encode information about the location of the objects and the feature vectors serve as input to the contrastive learning module which includes: 1) supervised contrastive learning for annotated positions, and 2) self-supervised debiased contrastive learning for unlabeled regions.

**a**

### EMPIAR-10304

| | crYOLO-3D | Template Matching | MiLoPYP |
|---|---|---|---|
| Precision | 0.475 | 0.589 | **0.826** |
| Recall | 0.146 | 0.655 | **0.856** |
| F1 | 0.224 | 0.620 | **0.841** |
| Number of annotations | 360 | N/A | 230 |

**b**

### EMPIAR-10499

| | crYOLO-3D | Template Matching | MiLoPYP |
|---|---|---|---|
| Precision | 0.478 | 0.261 | **0.715** |
| Recall | 0.568 | 0.553 | **0.764** |
| F1 | 0.520 | 0.355 | **0.739** |
| Number of annotations | 1100 | N/A | 195 |

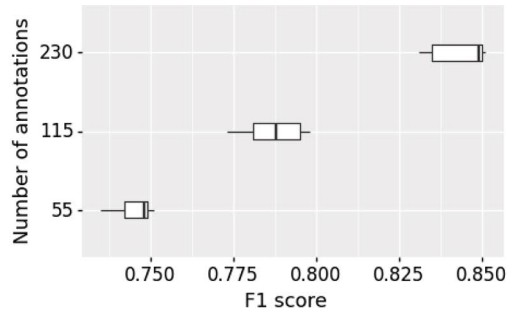

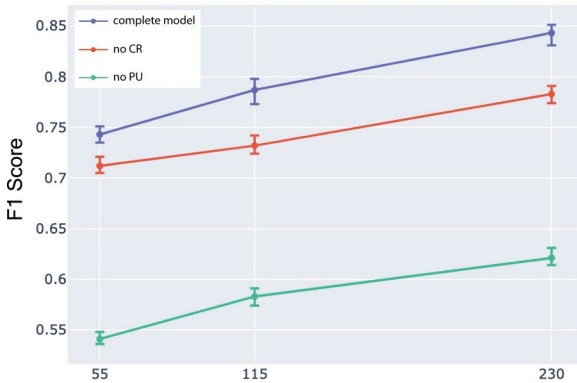

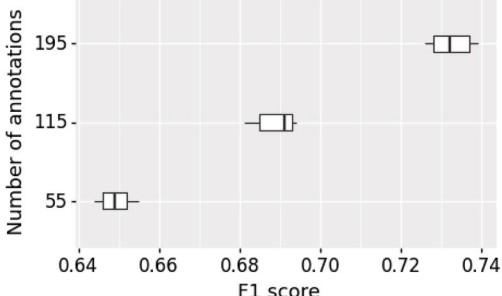

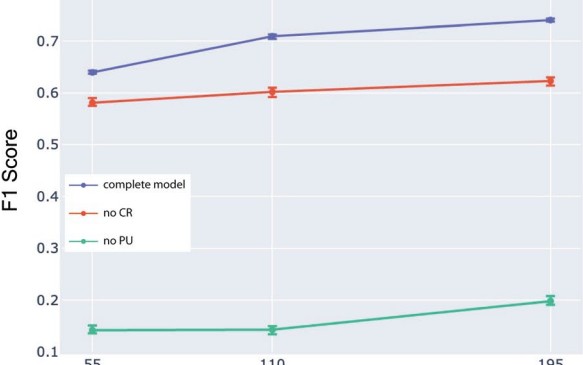

**Extended Data Fig. 3 | Detection performance and ablation studies on contrastive learning and positive unlabeled components.** Detection accuracy metrics were evaluated for in vitro and in situ datasets EMPIAR-10304 (**a**) and EMPIAR-10499 (**b**), respectively. Precision, recall and F1 scores were compared against crYOLO-3D and template matching (top row). We also measured detection performance using F1-scores when using increasing numbers of annotations during training (middle row). On the bottom row, we present mean, minimum and maximum average F1 scores for the 'complete model' including the contrastive (CR) and positive unlabeled (PU) learning components (blue), without the CR component (orange), and without the PU component (green), demonstrating the efficacy of the model. The upper bars represent the maximum minus the mean, and the lower bars represent the mean minus the minimum (sample size is 10 for EMPIAR-10304 and 5 for EMPIAR-10499).

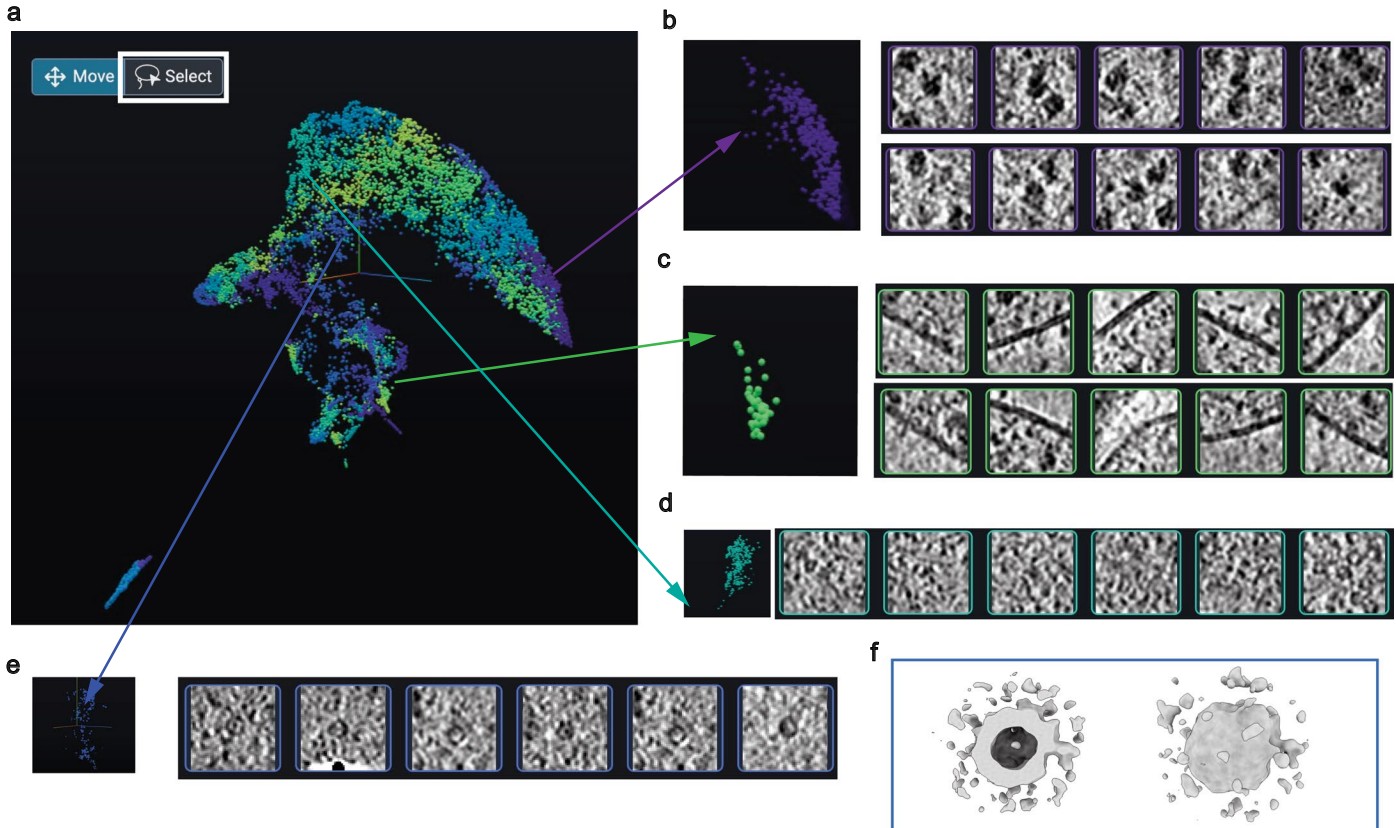

**Extended Data Fig. 4 | Interactive 3D visualization of tomogram embeddings from EMPIAR-10499. a**) 3D UMAP representation of protein embeddings colored according to labels assigned using over-clustering. Patches corresponding to clusters representing different species were isolated and selected using the 'select' button shown in a), including 70S ribosomes (**b**), membrane-like features (**c**), unidentified small-sized proteins (**d**), and circularly shaped vesicles (**e**). **f**) Low-resolution 3D reconstruction obtained from patches containing circularly shaped objects representing a hollow vesicle.

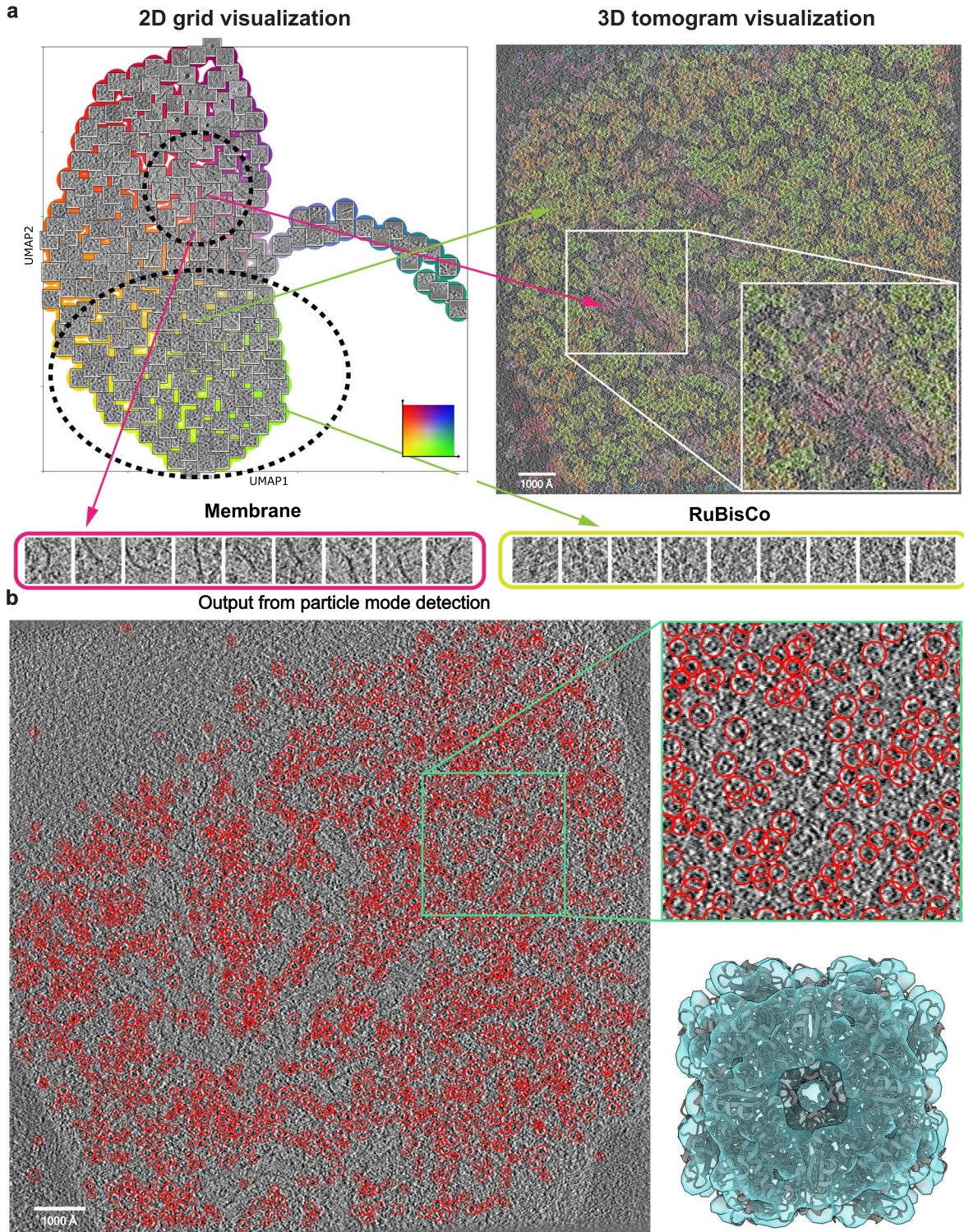

**Extended Data Fig. 5 | Detection and structure determination of RuBisCo enzymes from cellular tomograms. a**) 2D grid visualization of embeddings obtained from tilt series downloaded from EMPIAR-10694 (top, left). RuBisCo particles are seen as the most prominent feature (yellow/green). Membrane-like features (magenta) are also observed. Tomographic slice colored according to the embedding coordinates (top, right). A zoomed-in view highlighted in white is also provided. A selection of patches containing membrane (magenta) and RuBisCo particles (lime) are shown below. **b**) Annotated tomogram with detected RuBisCo particles colored in red (left). Zoomed-in view showing the accurate localization of RuBisCo particles (top, right). 11.0 Å resolution reconstruction of RuBisCo obtained from 35,352 particles with PDB coordinates 1GK8 fit into the map (bottom, right).

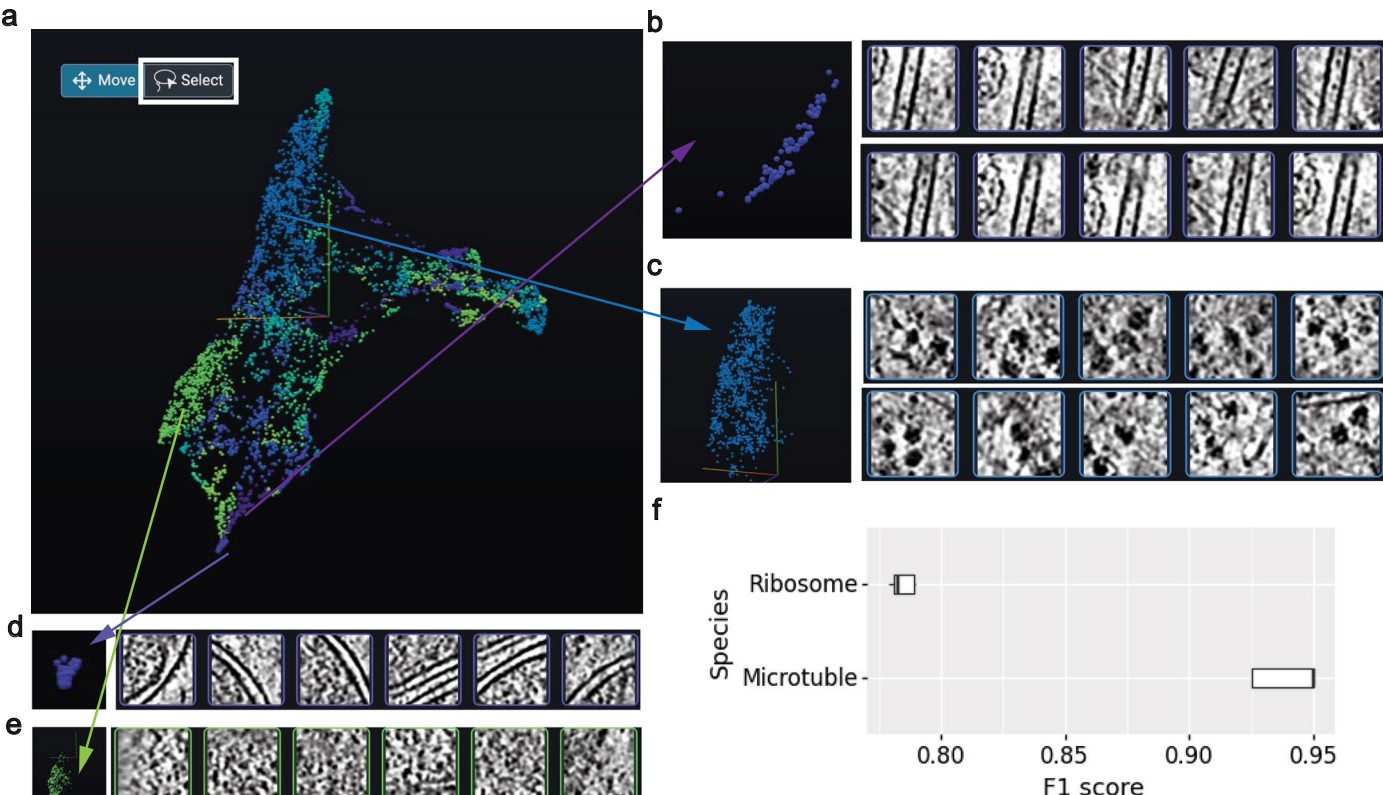

**Extended Data Fig. 6 | Interactive 3D visualization of tomogram embeddings from EMPIAR-10987. a**) 3D UMAP representation of protein embeddings colored according to labels assigned using over-clustering. Patches corresponding to clusters representing different species were isolated and selected using the 'select' button shown in a), including microtubules (**b**), 80S ribosomes (**c**), membrane-like features (**d**), and other small proteins

(**e**). **f**) Detection performance of ribosomes and microtubules measured using F1-scores. Each box extends from the first (Q1) to the third quartile (Q3). The median is marked by a line inside the box. Whisker lines correspond to box edges +/1.5 times the interquartile range (sample size is 10 for microtubules and 2 for ribosomes).

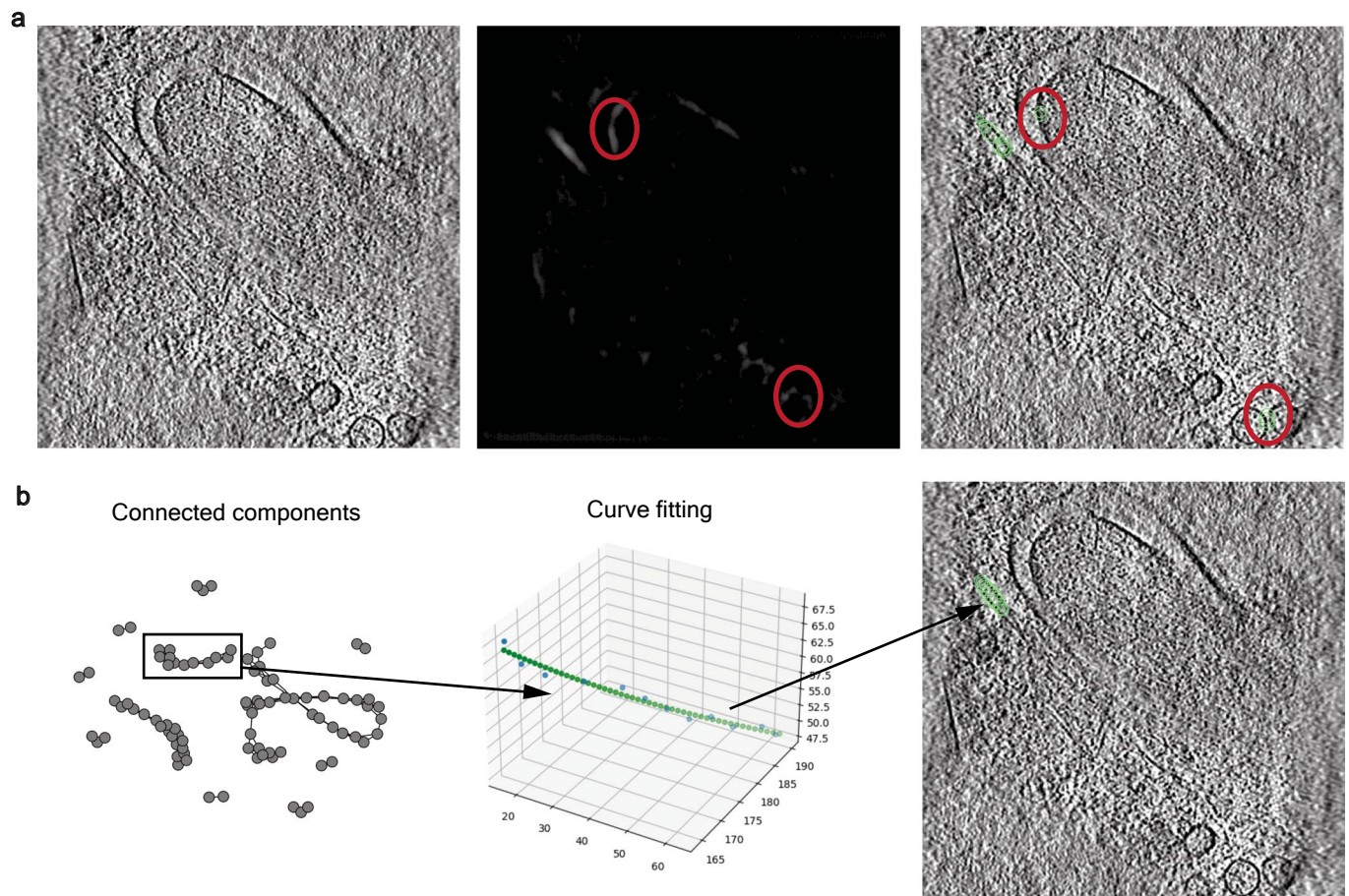

**Extended Data Fig. 7 | Post-processing pipeline for tubular-shaped protein detection. a**) Tomographic slice showing tubular structures observed in tilt series from EMPIAR-10987 obtained from in situ lamellae (left). Output heatmap from the particle localization module with bright areas representing detected microtubules (middle). Output coordinates (green circles) for detected particles prior to post-processing (right). False positive regions are highlighted in red. **b**)

The post-processing stage is composed of three steps: 1) convert 3D coordinates into a graph representation and identify connected components within the graph (left), 2) polynomial curve fitting for each connected component and evaluation of the quality of fit (middle), and 3) curves with fitting values above a user-defined threshold and with curvature smaller than a preset threshold are extracted as the final coordinates representing microtubules (right).

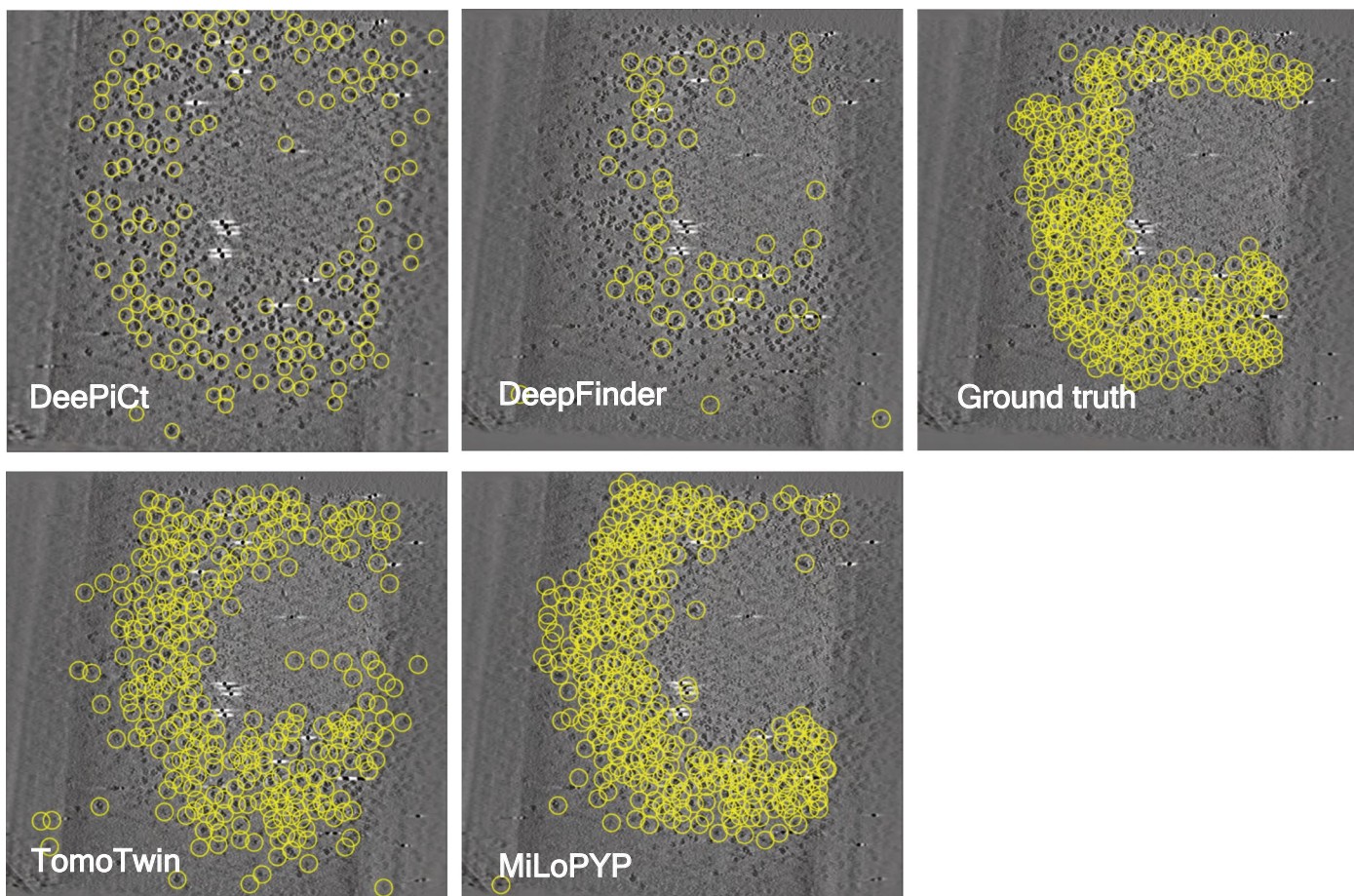

**Extended Data Fig. 8 | Comparison of picking performance between MiLoPYP and other deep-learning-based approaches on EMPIAR-10304.** Tomographic slices showing particles picked by DeePiCt, DeepFinder, TomoTwin, MiLoPYP, and corresponding ground truth positions on a tomogram with purified *E. coli* 70S ribosomes downloaded from EMPIAR-10304.

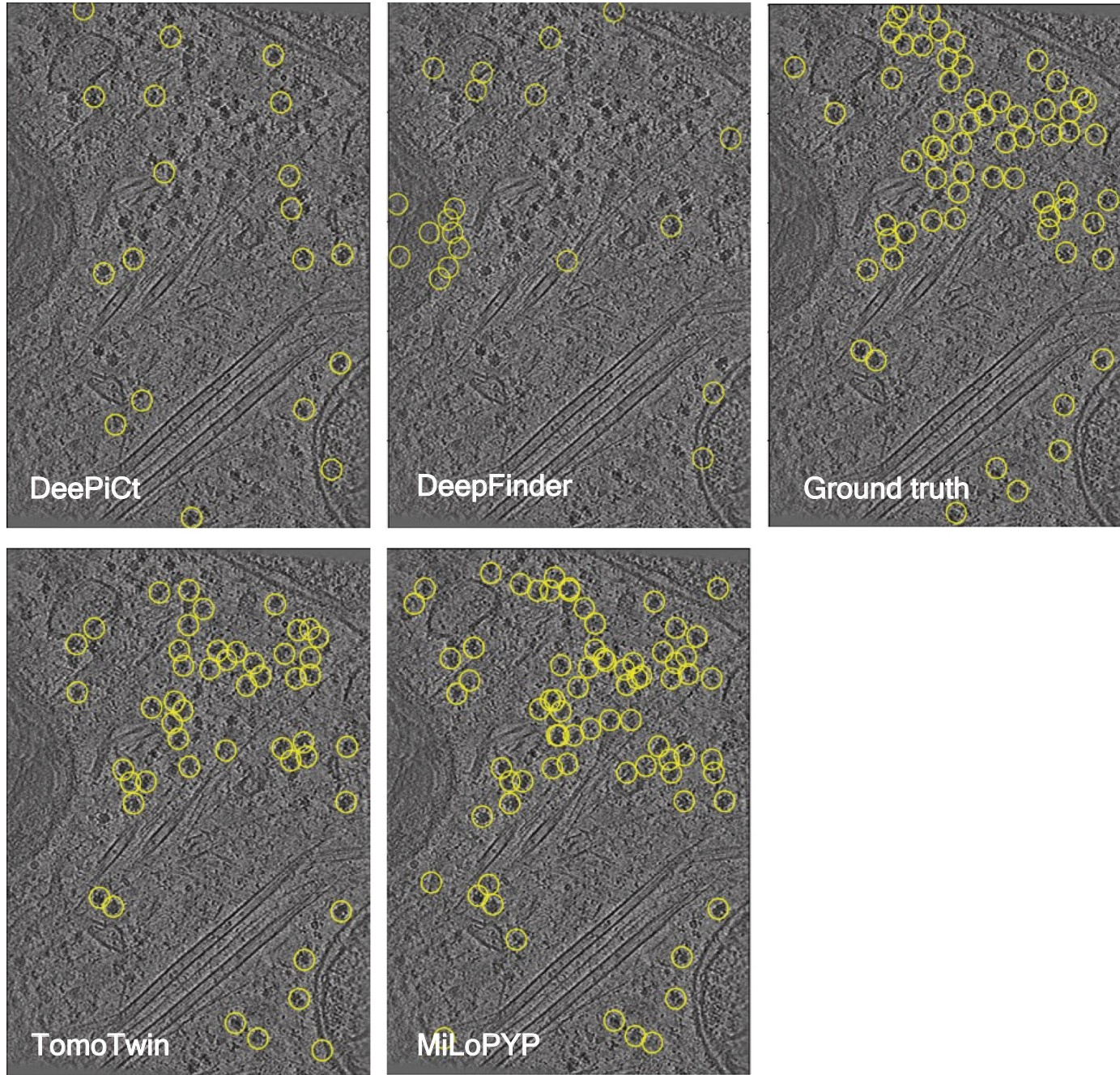

**Extended Data Fig. 9 | Comparison of picking performance between MiLoPYP and other deep-learning-based approaches on EMPIAR-10987.** Tomographic slices showing particles picked by DeePiCt, DeepFinder, TomoTwin, MiLoPYP and corresponding ground truth positions on a tomogram collected from thinned lamellae of mouse mammary gland epithelial EpH4 cells downloaded from EMPIAR-10987.

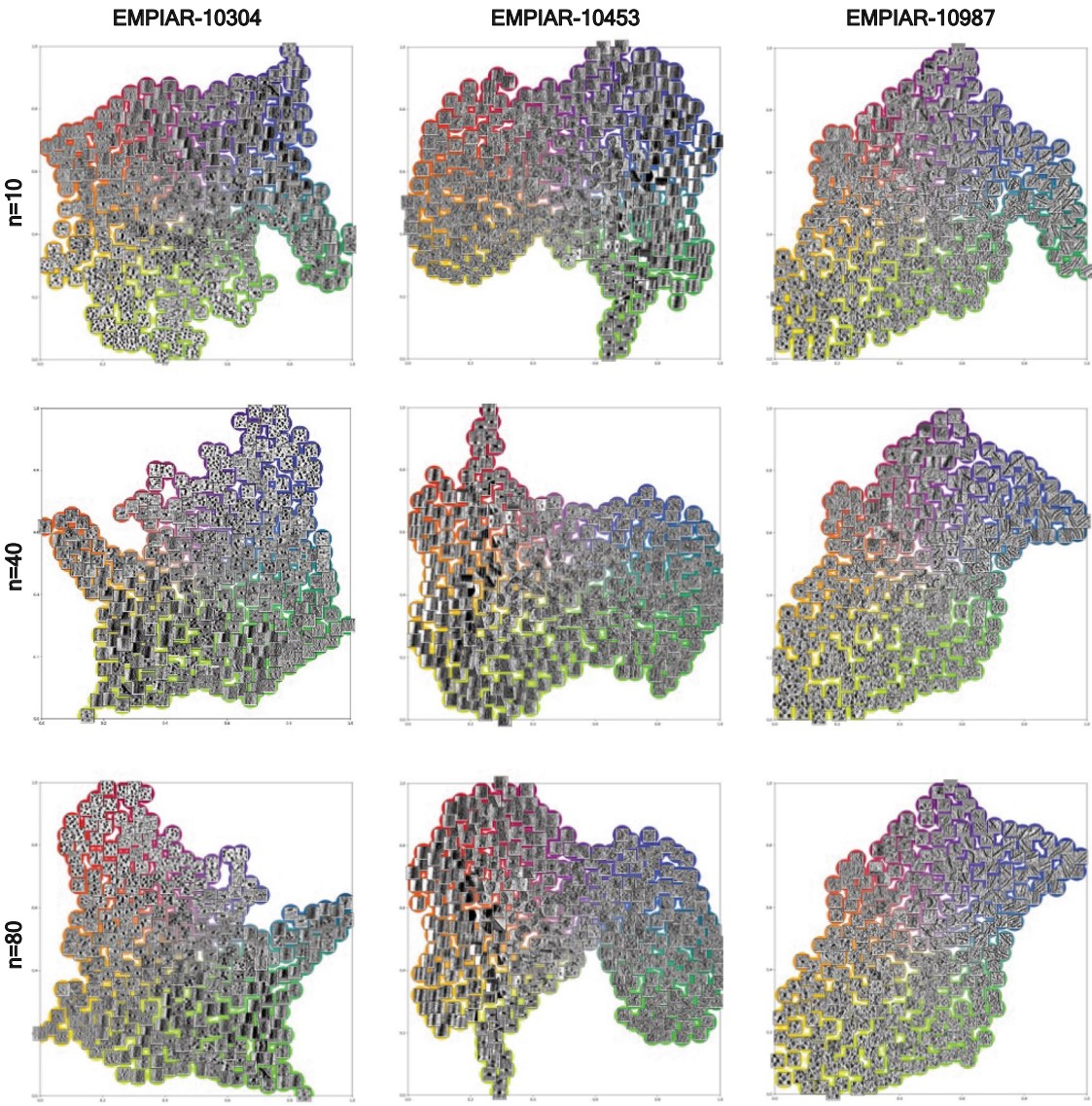

**Extended Data Fig. 10 | Effect of the number of neighbors used in the UMAP representation during the exploration phase.** Panels show changes in the UMAP representation as a function of the number of neighbors (10, 40 and 80). Results are shown for EMPIAR-10304, EMPIAR-10453 and EMPIAR-10987. The use of fewer neighbors reveals finer local structure while the use of more neighbors captures the overall structure with some loss of fine local structure. However, the full vector representation for each sub-volume remains unchanged, as UMAP only serves as a post-processing dimensionality reduction technique for the purpose of visualization.

# Reporting Summary

## Statistics

For all statistical analyses, confirm that the following items are present in the figure legend, table legend, main text, or Methods section.

| n/a | Confirmed | |
|---|---|---|
| ☐ | ☒ | The exact sample size (*n*) for each experimental group/condition, given as a discrete number and unit of measurement |
| ☐ | ☒ | A statement on whether measurements were taken from distinct samples or whether the same sample was measured repeatedly |
| ☒ | ☐ | The statistical test(s) used AND whether they are one- or two-sided<br>*Only common tests should be described solely by name; describe more complex techniques in the Methods section.* |
| ☒ | ☐ | A description of all covariates tested |
| ☒ | ☐ | A description of any assumptions or corrections, such as tests of normality and adjustment for multiple comparisons |
| ☐ | ☒ | A full description of the statistical parameters including central tendency (e.g. means) or other basic estimates (e.g. regression coefficient) AND variation (e.g. standard deviation) or associated estimates of uncertainty (e.g. confidence intervals) |
| ☒ | ☐ | For null hypothesis testing, the test statistic (e.g. *F*, *t*, *r*) with confidence intervals, effect sizes, degrees of freedom and *P* value noted<br>*Give P values as exact values whenever suitable.* |
| ☒ | ☐ | For Bayesian analysis, information on the choice of priors and Markov chain Monte Carlo settings |
| ☐ | ☒ | For hierarchical and complex designs, identification of the appropriate level for tests and full reporting of outcomes |
| ☒ | ☐ | Estimates of effect sizes (e.g. Cohen's *d*, Pearson's *r*), indicating how they were calculated |

*Our web collection on statistics for biologists contains articles on many of the points above.*

## Software and code

Policy information about availability of computer code

| Data collection | No new data collection was performed for this study. |
|---|---|
| Data analysis | Publicly available software used for data analysis: nextPYP v0.6.5 (http://nextpyp.app), PyTorch v1.11.0, umap v0.5.3, and Phoenix v0.0.27. Custom software developed for this study: MiLoPYP v0.6.5 (https://github.com/nextpyp/cet_pick). |

For manuscripts utilizing custom algorithms or software that are central to the research but not yet described in published literature, software must be made available to editors and reviewers. We strongly encourage code deposition in a community repository (e.g. GitHub). See the Nature Portfolio guidelines for submitting code & software for further information.

## Data

Policy information about availability of data

All manuscripts must include a data availability statement. This statement should provide the following information, where applicable:
- Accession codes, unique identifiers, or web links for publicly available datasets
- A description of any restrictions on data availability
- For clinical datasets or third party data, please ensure that the statement adheres to our policy

This study utilized raw tilt-series available from the Electron Microscopy Public Image Archive (EMPIAR) database under accession numbers 10304, 10499, 10453, 10987, and 11658 and cryo-EM maps available from the Electron Microscopy Data Bank (EMDB) under accession numbers 23358, 11650, 11347, 10061 and 33118. Cryo-EM density maps produced in this study were deposited in the EMDB under accession nos. EMD-45261 for EMPIAR-10304, EMD-45266 for EMPIAR-10499,

EMD-45267 and EMD-45268 for EMPIAR-10453, EMD-45269 for EMPIAR-10694, EMD-45270 and EMD-45271 for EMPIAR-10987, and EMD-45272 for EMPIAR-11658.

## Human research participants

Policy information about studies involving human research participants and Sex and Gender in Research.

| | |
|---|---|
| Reporting on sex and gender | No Human research participants were used in this study. |
| Population characteristics | N/A |
| Recruitment | N/A |
| Ethics oversight | N/A |

Note that full information on the approval of the study protocol must also be provided in the manuscript.

# Field-specific reporting

Please select the one below that is the best fit for your research. If you are not sure, read the appropriate sections before making your selection.

☒ Life sciences          ☐ Behavioural & social sciences          ☐ Ecological, evolutionary & environmental sciences

For a reference copy of the document with all sections, see nature.com/documents/nr-reporting-summary-flat.pdf

# Life sciences study design

All studies must disclose on these points even when the disclosure is negative.

| | |
|---|---|
| Sample size | All raw data used in this study were obtained from datasets deposited in the EMPIAR database. The total number of tilt-series available for each entry was used, as follows: EMPIAR-10304 (12 tilt-series), EMPIAR-10499 (65 tilt-series), EMPIAR-10453 (266 tilt-series), EMPIAR-10987 (20 tilt-series), and EMPIAR-11658 (260 tilt-series). Datasets were selected to represent a wide variety of in vitro and in situ cryo-ET samples, as needed to fully evaluate the performance of MiLoPYP. |
| Data exclusions | All data from the corresponding entries were used for processing. No data were excluded from the analyses. |
| Replication | Detailed protocols for obtaining the final maps were documented and used by at least three different lab members to reproduce the results. Replication attempts resulted in similar embeddings, same number of particles, and maps with identical resolutions. |
| Randomization | Randomization was not relevant in this study. All tilt-series were used for the analyses so no randomization was needed. |
| Blinding | Blinding was not relevant for this study because all tilt-series were used for the analyses. No data were left out. |

# Reporting for specific materials, systems and methods

We require information from authors about some types of materials, experimental systems and methods used in many studies. Here, indicate whether each material, system or method listed is relevant to your study. If you are not sure if a list item applies to your research, read the appropriate section before selecting a response.

## Materials & experimental systems

| n/a | Involved in the study |
|---|---|
| ☒ | ☐ Antibodies |
| ☒ | ☐ Eukaryotic cell lines |
| ☒ | ☐ Palaeontology and archaeology |
| ☒ | ☐ Animals and other organisms |
| ☒ | ☐ Clinical data |
| ☒ | ☐ Dual use research of concern |

## Methods

| n/a | Involved in the study |
|---|---|
| ☒ | ☐ ChIP-seq |
| ☒ | ☐ Flow cytometry |
| ☒ | ☐ MRI-based neuroimaging |

