## [Peer Review File · Nature Methods]

MiLoPYP: self-supervised molecular pattern mining and particle localization in situ

Corresponding Author: Dr Alberto Bartesaghi

Version 0:

Decision Letter:

7th Feb 2024

Dear Alberto,

Your Article entitled "MiLoPYP: self-supervised molecular pattern mining and particle localization in situ" has now been seen by three reviewers, whose comments are attached. While they find your work of potential interest, they have raised serious concerns which in our view are sufficiently important that they preclude publication of the work in Nature Methods, at least in its present form.

As you will see, the reviewers raise concerns about the benefits of the method over existing tools and its general applicability to challenging intracellular targets (and true membrane proteins).

Should further experimental data allow you to fully address these criticisms we would be willing to look at a revised manuscript (unless, of course, something similar has by then been accepted at Nature Methods or appeared elsewhere). This includes submission or publication of a portion of this work somewhere else. We hope you understand that until we have read the revised paper in its entirety we cannot promise that it will be sent back for peer-review.

Although we cannot publish your paper, it may be appropriate for another journal in the Nature Portfolio. If you wish to explore the journals and transfer your manuscript please use our manuscript transfer portal. You will not have to re-supply manuscript metadata and files, unless you wish to make modifications. For more information, please see our [manuscript transfer FAQ](http://www.nature.com/authors/author_resources/transfer_manuscripts.html?WT.mc_id=EMI_NPG_1511_AUTHORTRANSF&WT.ec_id=AUTHOR) page.

If you are interested in revising this manuscript for submission to Nature Methods in the future, please contact me to discuss your appeal before making any revisions. Otherwise, we hope that you find the reviewers' comments helpful when preparing your paper for submission elsewhere.

Sincerely,
Rita

Rita Strack, Ph.D.
Senior Editor
Nature Methods

Reviewers' Comments:

Reviewer #1:
Remarks to the Author:
Remarks to the authors

Summary

The manuscript introduces a stand-alone software package, MiLoPYP, that has also been integrated into the NextPYP platform. MiLoPYP consists of two modules: cellular mining of tomograms and particle localization for subtomogram analysis / single particle tomography. The cellular mining module is based on a Difference of Gaussians pyramid, a feature enhancement that compares to a number of spatial band-pass filters, to identify crucial coordinates in the tomogram. Those coordinates are

subsequently used to extract subtomograms for contrastive representation learning with a Siamese network. Via this method, the authors demonstrate that the trained network can be used as a data exploration tool that maps similar subtomograms close to each other in a high dimensional space. Huang & Zhou et al. then go on to demonstrate a semi-supervised particle picking workflow for localising particles identified using the data exploration tool. The authors use the platform for the localization and averaging of particles ranging from ribosomes to microtubules to spike proteins. Huang et al. demonstrate that their algorithms can detect free-floating spike proteins as well as its membrane-attached version in the EMPIAR-10453 dataset and reconstruct both ribosomes and microtubules from cellular tomograms collected on lamellae.

Overall, the MiLoPYP package is a technically sound tool and holds potential for the cryo-ET community. I believe it will be of interest to the broad readership of Nature Methods. Below, I have assembled a number of concerns I have with the presentation and contextualisation of the tool within the field of cryo-ET data analysis that I believe would strengthen the claim of MiLoPYP as a "general purpose tool for in situ pattern mining".

Major comments

1. While the exact implementation is new to the field of cryoET and exciting, the comparison to the start-of-the-art could be improved in my opinion. Currently, MiLoPYP is only being compared to the crYOLO3D pipeline and the classical template matching approach. Benchmarking of MiLoPYP against software packages with similar functionalities such as DeepFinder (particle localization), TomoTwin (Mining and particle localization via reference-based or clustering-based picking), and DeePiCt (particle localization) would be useful to be able to judge where MiLoPYP stands in comparison to the state-of-the-art.

2. Regarding particle localization, I agree that sparse labels will be less time consuming than the previously described fully supervised approaches. That being said, it would still be useful to see the comparison of F1 scores between MiLoPYP and other approaches (DeepFinder, DeePiCt, TomoTwin) when using the same amount of labels. In our experience, using data augmentation even relatively small number of labels can give good results for particle picking using CNNs. From the current manuscript, it is not becoming clear to me whether the performance would be similar or even improved with MiLoPYP's semi-supervised refinement approach. A comparison to these tools using for example the synthetic SHREC dataset (compare Deepfinder paper, Extended Data Fig. 4) and a some of the EMPIAR datasets presented throughout the manuscript would be helpful.

3. In the definition of its name, they authors imply MiLoPYP to be useful for cellular mining (a "dataset-specific cellular pattern Mining (Mi) and particle Localization (Lo) Python (PY) Pipeline (P)"). The M. pneumonia dataset contains more than ribosomes and membranes. Same goes for other available large datasets collected on lamellae (e.g. EMPIAR-11658) that could be used to demonstrate the application of MiLoPYP. It would greatly strengthen the manuscript to demonstrate that macromolecules other than the highest-contrast protein complexes in cellular tomography (microtubule, ribosomes) can be localised and resolved in cellular datasets. Supplementary Figure 4 is going into that direction already, so expanding a similar analysis to the larger yeast dataset (EMPIAR-11658) should enable to demonstrate the power of MiLoPYP as a tool for cellular mining.

4. Huang & Zhou et al. state that the usage of "only 55 annotations [of ribosomes] produced a high detection F1 of 0.73, demonstrating MiLoPYP's ability to locate targets present at low concentrations". Ribosomes, however, are not targets present at low concentration and later in the discussion, the class imbalance problem is mentioned that will render rare proteins difficult to detect. Thus, the authors only demonstrates here that few annotations are needed to achieve decent results via the semi-supervised refinement approach for highly abundant proteins. Whether particles at low concentration can be identified remains elusive.

5. Regarding the application to EMPIAR-10453, how many free-floating spikes are falsely detected in the membrane-attached spikes when inference is run on the entire dataset? In general, I could not find any performance statistics (F1 scores in comparison to other tools etc.) on EMPIAR-10453 throughout the entire manuscript. I apologize if I simply missed it but otherwise, I would advise the authors to include this in the manuscript.

6. The abstract states that "native membrane proteins" can be detected, but later on in the introduction and throughout the main text it is referred to as "membrane-attached proteins". The authors may consider to tone this down in the abstract and results section as well. Also, it is stated in the discussion section that MiLoPYP "can accurately detect membrane-bound and tubular-shaped complexes, converting it into a versatile and general purpose tool for in situ molecular pattern mining". The demonstration for membrane-bound complexes was, however, only in the best case scenario regarding contrast, isolated viral particles. The question remains whether this would be possible in the cellular context, which would be the requirement for a "general purpose tool for in situ pattern mining". I recommend to weaken the claim, as membrane proteins generally remain challenging for cryo-ET.

Minor comments

1. I. 37: A great recent citation for template matching would also be the GPU implementation of PyTOM (<https://doi.org/10.3390/ijms241713375>) and PyTME (<https://doi.org/10.1016/j.softx.2024.101636>)

2. I. 151-152 It sounds like the tubular-shaped complexes are imaged by FIB/SEM volume imaging applications. While obvious to the expert, I would consider rephrasing to make clear this indeed cryo-ET on FIB-milled lamellae.

3. The DoG-CB3D paper that assessed Difference-of-Gaussians for reference-free particle picking should be included.

Reviewer #2:

Remarks to the Author:

MiLoPYP is a two-step contrastive learning framework for mining and localization of cellular structures in situ. The first step of MiLoPYP is aimed at learning a representation/embedding space where similar particles are grouped together while dissimilar ones are separated. Since annotation is unavailable at this stage, positive pairs are generated via data augmentation, which follows the way of SimCLR, a well-known contrastive learning framework in the field of computer vision. Once trained, MiLoPYP allows users to manually select clusters or regions in the embedding spaces with the aid of three visualization techniques. Each cluster contains embeddings mapped with positions in the tomogram where a particle may occupy. The second step is to refine the coordinates with a semi-supervised detection manner, which yields a heatmap for post-processing. The detection framework implements a CenterNet-based network for 3D tomograms. According to the results reported in this manuscript, MiLoPYP achieves a good performance on given experimental data, but more tests are still needed to demonstrate its performance on more general cases, such as dense and relatively-small proteins in cell.

Major concerns:

- 1) The Gaussian Difference (DoG) method is used to extract the coarse coordinates of the target in this work. Is it required that the coordinates of coarse recognition should be accurate sufficiently, and how much impact will it have on the later learning process if there is a large error? And is the DoG recognition method reliable when the tomogram quality is poor or the particle distribution is very dense?
- 2) The training of the embedding network uses two types of information: low-angle projection photos and center sections 3D tomograms as inputs. Is this method better than direct input of 3D particles from tomogram? In addition, if the sample contains a large number of other objects in the z-direction, these objects will be severely aliased in the projection micrograph, will this have a negative effect on the training of the network?
- 3) The authors used spikes on the surface of the SARS-CoV-2 virus to test membrane proteins, but this does not indicate the ability to recognize membrane proteins. Although the spike here is a membrane protein, most of the protein is outside the membrane, and the extramembrane region has a very strong image signature. However, most membrane proteins are only a small fraction outside the membrane, which is usually difficult to recognize and process. Therefore, it is very misleading to claim that the software can recognize membrane proteins.
- 4) Similar to the previous question, it is not enough to use the identification of ribosomes to illustrate the performance of the software. Ribosome is now the most studied protein using cryoET methods, mainly because of its large molecular weight, strong intracellular contrast, and lack of crowding. Therefore, the authors should not just use this simple example for recognition to demonstrate performance, but should try to identify proteins that are densely present in cells with only a few hundred KDa.
- 5) Can the method of pre-trained feature extraction network used in this paper be compared with TomoTwin? MiLoPYP allows users to select clusters via visualization such as 2D grid method. Tomotwin introduces a similar clustering workflow. The difference is that MiLoPYP adds a refinement step/network based on CenterNet-3D. To my comprehension, this is used for fixing the error/bias caused by DoG. Can you evaluate the gain of this refinement step?
- 6) How does MoLiPYP deal with missing wedge?

Reviewer #3:

Remarks to the Author:

In this paper, the authors propose a two-step dataset specific contrastive learning framework that can quickly mine molecular patterns and then perform accurate localization of proteins and other complexes. This work provides a general tool for comprehensive exploration of cellular content and accurate localization of proteins, with the ability to run fast and require minimal human intervention. The issues studied in this article are relatively innovative. However, there are still some serious issues with the motivation and technique of the manuscript, which must be solved before it is considered for publication.

The comments are summarized as follows.

Comment 1. In the introduction section, the authors mention DISCA, an unsupervised clustering-based method, and TomoTwin, a representation learning-based method, and point out their limitations in solving the molecular pattern mining problem. If the authors could compare the proposed model MiLoPYP with these two methods, perhaps it would make the model more convincing.

Comment 2. In "Accurate detection and localization of proteins imaged in vitro" part, a cluster containing 230 particles is selected to train the module for protein localization. It is desirable for the authors to explain the selection mechanism of the training set here, so as to enlighten potential model users in the future.

Comment 3. The topic of this paper is molecular pattern mining and particle localization. Due to the complexity and diversity of particle distribution within the cell, we expected the authors to use one or two other datasets for the task of protein or other particle localization to more convincingly demonstrate the validity of the model.

Comment 4. In the Online Methods section, the author uses several random enhancement in the contrastive learning module to maintain instance identity and promote effective learning. But this usually makes the convergence of the model worse. Therefore, we hope that the authors will demonstrate the necessity of incorporating these training methods through ablation experiments.

Comment 5. In the section 'Accurate detection and localization of proteins imaged in vitro', the author compares the proposed model with conventional template matching and the deep learning-based approach crYOLO-3D. However, the author did not give the parameter settings used when running the comparison method. The effect of deep learning methods often depends on the parameters used, so we hope that the authors can add explanations of the parameter values in the two comparison methods to show the fairness of the comparison.

Comment 6. There is a spelling error in "constrastive" on line 58, it should be "contrastive". For "time consuming" on line 64, a hyphen needs to be added between "time" and "consuming". "STA" on line 99 may need to be explained or given its full name to ensure that readers understand it.

Comment 7. In the discussion section, the author mentioned that the 2D visualization step in MiLoPYP is prone to class imbalance. Please analyze this quantitatively. For example, the authors could show changes in model performance when changing the number of neighbors in UMAP and tsne low-dimensional embeddings.

** For Nature Portfolio general information and news for authors, see <http://npg.nature.com/authors>.

Version 1:

Decision Letter:

12th Feb 2024

Dear Alberto,

Thank you for your letter asking us to reconsider our decision on your Article, "MiLoPYP: self-supervised molecular pattern mining and particle localization in situ". After careful consideration we have decided that we are willing to consider a revised version of your manuscript that contains the experiments you've proposed. Please note, we cannot make promises about sending the manuscript back for review until we've seen the results of these experiments.

- * include a point-by-point response to our referees and to any editorial suggestions
- * please underline/highlight any additions to the text or areas with other significant changes to facilitate review of the revised manuscript
- * address the points listed described below to conform to our open science requirements
- * ensure it complies with our general format requirements as set out in our guide to authors at www.nature.com/naturemethods
- * resubmit all the necessary files electronically by using the link below to access your home page

Link Redacted

We hope to receive your revised paper within three months. If you cannot send it within this time, please let us know. In this event, we will still be happy to reconsider your paper at a later date so long as nothing similar has been accepted for publication at Nature Methods or published elsewhere.

OPEN SCIENCE REQUIREMENTS

REPORTING SUMMARY AND EDITORIAL POLICY CHECKLISTS

When revising your manuscript, please submit reporting summary and editorial policy checklists.

DATA AVAILABILITY

CODE AVAILABILITY

Please include a "Code Availability" subsection in the Online Methods which details how your custom code is made available. Only in rare cases (where code is not central to the main conclusions of the paper) is the statement "available upon request" allowed (and reasons should be specified).

MATERIALS AVAILABILITY

ORCID

Nature Methods is committed to improving transparency in authorship. As part of our efforts in this direction, we are now requesting that all authors identified as 'corresponding author' on published papers create and link their Open Researcher and Contributor Identifier (ORCID) with their account on the Manuscript Tracking System (MTS), prior to acceptance. This applies to primary research papers only. ORCID helps the scientific community achieve unambiguous attribution of all scholarly contributions. You can create and link your ORCID from the home page of the MTS by clicking on 'Modify my Springer Nature account'. For more information please visit <http://www.springernature.com/orcid>.

Sincerely,
Rita

Rita Strack, Ph.D.
Senior Editor
Nature Methods

Author Rebuttal letter:

We thank the reviewers for evaluating our work and for the constructive feedback which we have incorporated into the revised version of our manuscript.

Here is a summary of the major changes:

- We added the analysis of a new cellular dataset from *S. cerevisiae* yeast cells (EMPIAR-11658) and showed that MiLoPYP can successfully detect and localize ATP synthase complexes located on mitochondrial cristae (Figure 6).
 - We analyzed a new dataset containing a “dense and a relatively small protein target”, the RuBisCo enzyme (560kDa) from *Chlamydomonas reinhardtii* cells (EMPIAR-10694), showing that MiLoPYP can correctly detect and localize these particles (Supp. Figure 5).
 - We added new experiments to compare the performance of MiLoPYP with state-of-the-art approaches (DeepFinder, DeePiCt, and TomoTwin), using the synthetic SHREC-21 dataset and 3 EMPIAR datasets (Supp. Table 1 and Supp. Figures 9 and 10).
 - We added new experiments to quantify the contribution of the refinement module to overall picking accuracy, in terms of Precision, Accuracy and F1 scores (Supp. Table 2).
 - We added new experiments to quantify the effect of the number of neighbors in the UMAP mappings used for visualization in the exploration module (Supp. Figure 8).
- Our point-to-point responses are included below (highlighted in blue).

Reviewers' Comments:

Reviewer #1:

Remarks to the Author:

Remarks to the authors

Summary

The manuscript introduces a stand-alone software package, MiLoPYP, that has also been integrated into the NextPYP platform. MiLoPYP consists of two modules: cellular mining of tomograms and particle localization for subtomogram analysis / single particle tomography. The cellular mining module is based on a Difference of Gaussians pyramid, a feature enhancement that compares to a number of spatial band-pass filters, to identify crucial coordinates in the tomogram. Those coordinates are subsequently used to extract subtomograms for contrastive representation learning with a Siamese network. Via this method, the authors demonstrate that the trained network can be used as a data exploration tool that maps similar subtomograms close to each other in a high dimensional space. Huang & Zhou et al. then go on to demonstrate a semi-supervised particle picking workflow for localising particles identified using the data exploration tool. The authors use the platform for the localization and averaging of particles ranging from ribosomes to microtubules to spike proteins. Huang et al. demonstrate that their algorithms can detect free-floating spike proteins as well as its membrane-attached version in the EMPIAR-10453 dataset and reconstruct both ribosomes and microtubules from cellular tomograms collected on lamellae.

Overall, the MiLoPYP package is a technically sound tool and holds potential for the cryo-ET community. I believe it will be of interest to the broad readership of Nature Methods. Below, I have assembled a number of concerns I have with the presentation and contextualisation of the tool within the field of cryo-ET data analysis that I believe would strengthen the claim of MiLoPYP as a “general purpose tool for in situ pattern mining”.

We thank the reviewer for the positive comments. In summary, we have strengthened the claim that MiLoPYP is a “general purpose tool for in situ pattern mining” by adding the analysis of two new datasets and by benchmarking our approach against the state-of-the-art methods in the field.

Major comments

1. While the exact implementation is new to the field of cryoET and exciting, the comparison to the start-of-the-art could be improved in my opinion. Currently, MiLoPYP is only being compared to the crYOLO3D pipeline and the classical template matching approach. Benchmarking of MiLoPYP against software packages with similar functionalities such as DeepFinder (particle localization), TomoTwin (Mining and particle localization via reference-based or clustering-based picking), and DeePiCt (particle localization) would be useful to be able to judge where MiLoPYP stands in comparison to the state-of-the-art.

We carried out new experiments to compare the performance of MiLoPYP against TomoTwin,

DeepFinder, and DeePiCt. To ensure thorough and meaningful comparisons, we used tomograms from four different datasets representing synthetic benchmarks (SHREC-21) and real datasets from in vitro and in situ samples (EMPIAR-10304: purified Ecoli 70S ribosomes, EMPIAR-10453: SARS-CoV-2 virus particles attached to VLPs, and EMPIAR-10987: ribosomes from Mouse mammary gland epithelial EpH4 cells). Particle localization performance was evaluated quantitatively in terms of F1 scores (Supp. Table 1 and Supp. Figures 9 and 10).

When evaluating TomoTwin, we used the latest pretrained model weights available from: <https://zenodo.org/records/8358240> (tomotwin_latest.pth) and followed the instructions from the “reference-based particle picking tutorial” (<https://tomotwin-cryoet.readthedocs.io/>).

For DeepFinder, F1 numbers for the SHREC-21 dataset were obtained from the SHREC-21 challenge manuscript (<https://arxiv.org/pdf/2203.10035.pdf>). When analyzing the EMPIAR datasets containing ribosomes (10304 and 10987), we used the pre-trained model weights (net_weights_chlamydomonas.h5) and followed the suggested tutorial instructions (<https://cryoet-deepfinder.readthedocs.io/>). Since the model was trained to detect 4 classes, the F1 numbers reported in Supplementary Table 1 correspond to the ribosome class only.

For DeePiCt, we used the pre-trained model provided by the authors (full_vpp_ribo_model_IF4_D2_BN.pth) and followed the tutorial instructions (<https://github.com/ZauggGroup/DeePiCt>) to detect ribosomes from EMPIAR-10304 and EMPIAR-10987. Since DeePiCt was only trained to detect ribosomes, membranes, microtubules and fatty acid synthase (FAS), we could not apply this method to the SHREC-21 dataset (F1 scores in this case are reported as N/A).

Also, since neither DeepFinder nor DeePiCt were trained to detect native viral spikes, we did not obtain meaningful results when analyzing tomograms from SARS-CoV-2 virus particles (EMPIAR-10453) using these methods, so F1 scores in these cases are also reported as N/A.

As expected from training with fewer labels (TomoTwin, MiLoPYP) and semi-supervised learning approaches (MiLoPYP), these methods showed a drop in F1 scores on the SHREC-21 dataset when compared to the fully-supervised method DeepFinder. In particular, low-molecular weight targets with highly asymmetric shapes were the hardest for MiLoPYP and TomoTwin to detect, while F1 scores for the larger complexes were similar to those obtained by DeepFinder. In contrast, performance for MiLoPYP and TomoTwin was significantly higher than DeepFinder and DeePiCt, both for the in vitro (EMPIAR-10304) and in situ (EMPIAR-10987) ribosome datasets, with MiLoPYP outperforming TomoTwin in terms of F1 scores by ~7% in both cases.

In summary, our experiments demonstrate that MiLoPYP compares favorably against state-of-the-art approaches for unsupervised data mining (TomoTwin) and fully-supervised particle picking (DeepPicker and DeePiCt), when evaluated on challenging tomograms both from in vitro and in situ samples (Supp. Table 1 and Supp. Figures 9 and 10).

2. Regarding particle localization, I agree that sparse labels will be less time consuming than the previously described fully supervised approaches. That being said, it would still be useful to see the comparison of F1 scores between MiLoPYP and other approaches (DeepFinder, DeePiCt, TomoTwin) when using the same amount of labels. In our experience, using data augmentation even relatively small number of labels can give good results for particle picking using CNNs. From the current manuscript, it is not becoming clear to me whether the performance would be similar or even improved with MiLoPYP's semi-supervised refinement approach. A comparison to these tools using for example the synthetic SHREC dataset (compare Deepfinder paper, Extended Data Fig. 4) and a some of the EMPIAR datasets presented throughout the manuscript would be helpful.

As mentioned above, we now present comparisons (in terms of F1 scores) between MiLoPYP, DeepFinder, DeePiCt and TomoTwin, both on SHREC and EMPIAR datasets. These experiments, however, were done using pre-trained models provided by the authors of each program, so clearly not all of them used the same amount of labels during training.

DeepFinder and DeePiCt are both fully-supervised approaches for particle localization and as such require exhaustive labeling of tomograms. According to data presented in the corresponding manuscripts, DeepFinder needs at least 1,400 ribosome labels “to achieve a competitive result” (Moebel et al., 2021: Extended Data Fig. 5), while DeePiCt needs a minimum of 2,450 ribosomes for training to be successful (Teresa-Trueba et al., 2023: Supplementary Figure 2). TomoTwin, also a fully-supervised approach but for pattern mining, was trained using 120,000 particles which was “necessary to learn a generalized representation of 3D macromolecular shapes” (Rice et al., 2023). In contrast, the number of labels used by MiLoPYP is significantly lower, between 50-230 annotations (Supplementary Figure 3). Given the significant differences in approach that exist between all these methods (i.e., localization vs. mining, fully-supervised vs. semi-supervised learning, etc.), it is challenging to make meaningful comparisons when the same number of labels are used for training.

For example, when comparing the performance of MiLoPYP against DeepFinder and TomoTwin on the SHREC'21 datasets. The pretrained weights provided with DeepFinder were obtained using the SHREC'19 dataset, and when we applied this model to the SHREC'21 dataset, even on protein species that exist in both SHREC'19 and SHREC'21, the model performed very poorly with a median F1 score of below 0.1. We suspect this is due to a major shift in data distribution between SHREC'19 and SHREC'21, reflecting limitations in transferability of fully-supervised models. However, when retrained using the SHREC'21 dataset, DeepFinder was able to achieve a median F1 score of 0.83. Using the pretrained model (from 120,000 particles), TomoTwin achieved a lower median F1 score of 0.57 on SHREC'21 while MiLoPYP achieved an F1 score of 0.58 when trained using only an average of 500 labels, demonstrating MiLoPYP's ability to produce promising results even when trained with less data.

In general, when exhaustive ground-truth labels are available, fully-supervised methods are expected to outperform semi-supervised methods, simply because there is more training data available. The main advantage of semi-supervised methods is their potential to perform well in real world scenarios with unseen data where no ground truth exists and limited manual labeling is available or desirable. When enough labels are provided to semi-supervised algorithms (and if the positive/negative loss functions are applied), these algorithms can achieve the same performance as fully-supervised algorithms.

3. In the definition of its name, they authors imply MiLoPYP to be useful for cellular mining (a "dataset-specific cellular pattern Mining (Mi) and particle Localization (Lo) Python (PY) Pipeline (P)"). The M. pneumonia dataset contains more than ribosomes and membranes. Same goes for other available large datasets collected on lamellae (e.g. EMPIAR-11658) that could be used to demonstrate the application of MiLoPYP. It would greatly strengthen the manuscript to demonstrate that macromolecules other than the highest-contrast protein complexes in cellular tomography (microtubule, ribosomes) can be localised and resolved in cellular datasets. Supplementary Figure 4 is going into that direction already, so expanding a similar analysis to the larger yeast dataset (EMPIAR-11658) should enable to demonstrate the power of MiLoPYP as a tool for cellular mining.

Indeed, we did originally try to go after cellular structures of macromolecules –other than microtubules and ribosomes– as shown in the original Supp. Figure 4. The main challenge we faced was the low concentrations at which these targets were present –which compounded with the limited number of tomograms available– precluded their analysis by single-particle tomography. Following the reviewer's suggestion, we downloaded tilt-series from the larger EMPIAR-11658 dataset and analyzed them using MiLoPYP. Interestingly, we were able to confirm that the 2D embedding plots showed the presence of membrane-attached proteins consistent with the shape and dimensions of ATP synthase. Since the quality of input data in terms of precision and number of coordinates for training is especially important for challenging datasets like this, we selected a cluster of particles from the 3D interactive session and manually added 85 additional particles to obtain a total of 247 coordinates that were used for training the refinement module. Using the trained model, we detected 4,105 particles from the entire dataset and subjected them to 3D refinement by single-particle tomography resulting in a 13 Å resolution map of ATP synthase obtained from 2,577 clean particles. In summary, the new experiments demonstrate MiLoPYP's ability to accurately detect and localize challenging targets such as membrane proteins from cellular tomograms. The results are presented in the new Figure 6 and new sections were added to Results and Online Methods describing the analysis.

4. Huang & Zhou et al. state that the usage of "only 55 annotations [of ribosomes] produced a high detection F1 of 0.73, demonstrating MiLoPYP's ability to locate targets present at low concentrations". Ribosomes, however, are not targets present at low concentration and later in the discussion, the class imbalance problem is mentioned that will render rare proteins difficult to detect. Thus, the authors only demonstrates here that few annotations are needed to achieve decent results via the semi-supervised refinement approach for highly abundant proteins. Whether particles at low concentration can be identified remains elusive.

By "low concentration" we mean the concentrations observed in situ (as in EMPIAR-10987), which are lower than those observed in purified samples (as in EMPIAR-10304). The point about highly abundant proteins is addressed by our new results on EMPIAR-11658 where we show that MiLoPYP can successfully detect and localize rare targets such as ATP synthase from less than 250 annotations. Note that compared to highly abundant and large proteins such as ribosomes, the detection of low abundance and smaller targets may require comparatively more labels (consistent with earlier findings, Moebel et al., 2021). We revised the text to clarify these points.

5. Regarding the application to EMPIAR-10453, how many free-floating spikes are falsely detected in the membrane-attached spikes when inference is run on the entire dataset? In general, I could not find any performance statistics (F1 scores in comparison to other tools etc.)

on EMPIAR-10453 throughout the entire manuscript. I apologize if I simply missed it but otherwise, I would advise the authors to include this in the manuscript.

For a tomogram with 35-40 spikes identified, an average of 3-5 particles correspond to free floating spikes (false positives). The reviewer is right, we did not include performance statistics on EMPIAR-10453 in the original version of the manuscript. We now report F1 scores obtained by MiLoPYP on this dataset (as well as comparisons with TomoTwin) in the new Supp. Table 1.

6. The abstract states that “native membrane proteins” can be detected, but later on in the introduction and throughout the main text it is referred to as “membrane-attached proteins”. The authors may consider to tone this down in the abstract and results section as well. Also, it is stated in the discussion section that MiLoPYP “can accurately detect membrane-bound and tubular-shaped complexes, converting it into a versatile and general purpose tool for in situ molecular pattern mining”. The demonstration for membrane-bound complexes was, however, only in the best case scenario regarding contrast, isolated viral particles. The question remains whether this would be possible in the cellular context, which would be the requirement for a “general purpose tool for in situ pattern mining”. I recommend to weaken the claim, as membrane proteins generally remain challenging for cryo-ET.

As mentioned above, we now show that MiLoPYP can successfully detect and localize ATP synthase complexes on cellular membranes, providing stronger evidence to support the claim that MiLoPYP is a general purpose tool for in situ pattern mining. Even though this is an exciting and promising result, these types of targets are still challenging to analyze, so we still decided to weaken the general claim that MiLoPYP is a “general purpose tool for in situ pattern mining”.

Minor comments

1. I. 37: A great recent citation for template matching would also be the GPU implementation of PyTOM (<https://doi.org/10.3390/ijms241713375>) and PyTME (<https://doi.org/10.1016/j.softx.2024.101636>)

Thank you for pointing this out, we added these references to the revised manuscript.

2. I. 151-152 It sounds like the tubular-shaped complexes are imaged by FIB/SEM volume imaging applications. While obvious to the expert, I would consider rephrasing to make clear this indeed cryo-ET on FIB-milled lamellae.

Thank you, we have made this clarification in the revision.

3. The DoG-CB3D paper that assessed Difference-of-Gaussians for reference-free particle picking should be included. (<https://doi.org/10.1186/s12859-016-1283-3>)

We added this reference to the section “Overview of MiLoPYP's workflow”.

Reviewer #2:

Remarks to the Author:

MiLoPYP is a two-step contrastive learning framework for mining and localization of cellular structures in situ. The first step of MiLoPYP is aimed at learning a representation/embedding space where similar particles are grouped together while dissimilar ones are separated. Since annotation is unavailable at this stage, positive pairs are generated via data augmentation, which follows the way of SimCLR, a well-known contrastive learning framework in the field of computer vision. Once trained, MiLoPYP allows users to manually select clusters or regions in the embedding spaces with the aid of three visualization techniques. Each cluster contains embeddings mapped with positions in the tomogram where a particle may occupy. The second step is to refine the coordinates with a semi-supervised detection manner, which yields a heatmap for post-processing. The detection framework implements a CenterNet-based network for 3D tomograms. According to the results reported in this manuscript, MiLoPYP achieves a good performance on given experimental data, but more tests are still needed to demonstrate its performance on more general cases, such as dense and relatively-small proteins in cell. We thank the reviewer for the constructive feedback. As detailed in the responses below, we analyzed a new dataset containing an example of a “dense and relatively small protein target”: the ribulose-1,5-bisphosphate carboxylase-oxygenase (RuBisCo) from *Chlamydomonas reinhardtii* cells (EMPIAR-10694). The new experiments demonstrate MiLoPYP's ability to successfully detect and localize relatively small targets (~500kDa) in situ.

Major concerns:

1) The Gaussian Difference (DoG) method is used to extract the coarse coordinates of the target in this work. Is it required that the coordinates of coarse recognition should be accurate sufficiently, and how much impact will it have on the later learning process if there is a large error? And is the DoG recognition method reliable when the tomogram quality is poor or the

particle distribution is very dense?

The main reason we apply the DoG method is to reduce the number of positions analyzed by the exploration module (i.e., to avoid doing an exhaustive per-voxel analysis). If the location of relevant features is missed after application of the DoG filter, these will not be used during the exploration phase, but generally, we have found DoG recognition to be reliable even on poor quality datasets as the results rely mostly on the low resolution information present in the tomograms. When particle concentration is very high, however, care has to be taken when choosing the parameters of DoG filtering (standard deviation and non-max suppression radius) to make sure that nearby particles are selected (see response to point 4 below for an example of a densely distributed sample). We clarified this point in the Results section.

2) The training of the embedding network uses two types of information: low-angle projection photos and center sections 3D tomograms as inputs. Is this method better than direct input of 3D particles from tomogram? In addition, if the sample contains a large number of other objects in the z-direction, these objects will be severely aliased in the projection micrograph, will this have a negative effect on the training of the network?

Yes, the use of low-tilt projections and central tomogram sections gave us better results than using the 3D sub-tomograms, and also resulted in better computational efficiency. The use of low-tilt projections is beneficial because the 2D data is not affected by the missing wedge (unlike the 3D data in the sub-tomograms). We note that the 2D projections are obtained by averaging multiple tilted projections at each tomogram position, so the averaging itself has a regularization effect that reduces the effect of any “incoherent” background components that may be present in the sample. In addition, since our approach also uses the central sections from the sub-tomograms (which are not affected by the background), this further helps diminish this problem. In cases where the background is very strong, it is possible to just use the central slices (and ignore the 2D projections). We clarified this in the Online Methods section.

3) The authors used spikes on the surface of the SARS-CoV-2 virus to test membrane proteins, but this does not indicate the ability to recognize membrane proteins. Although the spike here is a membrane protein, most of the protein is outside the membrane, and the extramembrane region has a very strong image signature. However, most membrane proteins are only a small fraction outside the membrane, which is usually difficult to recognize and process. Therefore, it is very misleading to claim that the software can recognize membrane proteins.

While our new experiments on EMPIAR-11658 demonstrate that MiLoPYP can effectively recognize ATP synthase in cellular membranes, in general, membrane proteins—especially those where most of the density is inside the membrane—remain difficult to detect by MiLoPYP or any other method. Following the reviewer’s suggestion, we toned down the original statement and clarified that detection and localization of smaller membrane proteins remains challenging.

4) Similar to the previous question, it is not enough to use the identification of ribosomes to illustrate the performance of the software. Ribosome is now the most studied protein using cryoET methods, mainly because of its large molecular weight, strong intracellular contrast, and lack of crowding. Therefore, the authors should not just use this simple example for recognition to demonstrate performance, but should try to identify proteins that are densely present in cells with only a few hundred KDa.

Indeed, studying ribosomes in situ poses fewer technical challenges compared to less abundant or lower molecular weight targets. To address this point (as mentioned above and in the response to Reviewer 1), we added two new experiments with non-ribosome targets. First, we present the analysis of tomograms from *Chlamydomonas reinhardtii* cells containing densely packed RuBisCO particles (EMPIAR-10694). We were able to detect copies of this ~560kDa molecule in the 2D embedding plots produced by MiLoPYP and successfully localize particles using our refinement module. The resulting particles were subjected to single-particle tomography in nextyPYP and used to produce an 11 Å resolution reconstruction. Second, we analyzed tilt-series from EMPIAR-11658 and used MiLoPYP to detect and extract ATP synthase particles (600kDa) that were used to determine a 13 Å resolution structure. These experiments demonstrate MiLoPYP performance and applicability beyond the most commonly studied ribosome samples (Supp. Figure 5 and Figure 6).

5) Can the method of pre-trained feature extraction network used in this paper be compared with TomoTwin? MiLoPYP allows users to select clusters via visualization such as 2D grid method. Tomotwin introduces a similar clustering workflow. The difference is that MiLoPYP adds a refinement step/network based on CenterNet-3D. To my comprehension, this is used for fixing the error/bias caused by DoG. Can you evaluate the gain of this refinement step?

As mentioned in the response to Reviewer 1, we added new experiments to compare the performance of MiLoPYP against TomoTwin using multiple datasets. Our results show that

MiLoPYP results in better detection accuracy as measured by F1 scores (Supp. Table 1 and Supp. Figures 9 and 10). MiLoPYP was specifically designed as a two-stage framework with dedicated exploration and localization modules to improve the accuracy of each component. Compared to TomoTwin, it should be noted that our exploration module requires additional training for each dataset in order to obtain accurate clustering results. During the exploration phase, the goal is to mine the entire complement of sub-volumes extracted from tomograms, while the goal of the refinement stage is to focus on localizing one specific target. Indeed, the main goal of the DoG step is to select the most “interesting” areas of the tomogram, and while this may lead to some missed particles, the use of the refinement module downstream allows us to improve the picking accuracy.

To address the point about the impact of the refinement module on the accuracy of particle localization, we carried out new experiments to quantify these differences in terms of Precision, Recall and F1 scores. The results show that the refinement module results in significant performance gains (Supp. Table 2 and Online Methods).

6) How does MoLiPYP deal with missing wedge?

While MiLoPYP does not have an explicit strategy to deal with the missing wedge, we did make several design decisions to try to minimize the effect of distortions stemming from this problem. First, we use 2D particle projections extracted from the raw tilt-series –which are not affected by the missing wedge– as input to the training module. Second, we also use averages of central consecutive slices from each sub-volume in an effort to reduce the missing wedge effects. While these strategies do not completely avoid the problem, they are very effective at reducing it. We clarified these points in the Introduction section.

Reviewer #3:

Remarks to the Author:

In this paper, the authors propose a two-step dataset specific contrastive learning framework that can quickly mine molecular patterns and then perform accurate localization of proteins and other complexes. This work provides a general tool for comprehensive exploration of cellular content and accurate localization of proteins, with the ability to run fast and require minimal human intervention. The issues studied in this article are relatively innovative. However, there are still some serious issues with the motivation and technique of the manuscript, which must be solved before it is considered for publication.

We thank the reviewer for the positive and detailed feedback. Below, we provide point-by-point responses to address the concerns raised regarding motivation and the technique.

The comments are summarized as follows.

Comment 1. In the introduction section, the authors mention DISCA, an unsupervised clustering-based method, and TomoTwin, a representation learning-based method, and point out their limitations in solving the molecular pattern mining problem. If the authors could compare the proposed model MiLoPYP with these two methods, perhaps it would make the model more convincing.

As mentioned in the response to Reviewers 1 and 2, we carried out new experiments comparing the performance of MiLoPYP against TomoTwin on multiple datasets (new Supplementary Table 1, new Supplementary Figures 9 and 10). When we tried to compare MiLoPYP against DISCA, however, we ran into difficulties running the program due to the lack of documentation and the unavailability of pretrained models. DISCA is part of the AITom package which is available on Github, and while the manuscript states that “There are more than 20 tutorials provided in AITom for different cryo-ET analysis tasks”, we could not find one for DISCA. In addition, the DISCA manuscript (Zeng et al., 2023) states that: “The trained models, demo data, and other generated data are available in AITom”, but we could not find these models anywhere in the AITom repository. Therefore, since we were only able to make direct comparisons between MiLoPYP and TomoTwin, we decided to include additional details in the Introduction section highlighting the main differences between MiLoPYP and DISCA.

Comment 2. In “Accurate detection and localization of proteins imaged in vitro” part, a cluster containing 230 particles is selected to train the module for protein localization. It is desirable for the authors to explain the selection mechanism of the training set here, so as to enlighten potential model users in the future.

We have expanded the description on how the training set is selected to the following: “Using the 3D interactive session, we obtained a subset of particles that shared the same label (assigned through over-clustering). Within this subset, a cluster comprising 230 particles was selected by visual inspection of the cluster centroids (Online methods).”

Comment 3. The topic of this paper is molecular pattern mining and particle localization. Due to

the complexity and diversity of particle distribution within the cell, we expected the authors to use one or two other datasets for the task of protein or other particle localization to more convincingly demonstrate the validity of the model.

As mentioned above, we have included the analysis of two new EMPIAR datasets containing cellular tomograms from *S. cerevisiae* (EMPIAR-11658) and *Chlamydomonas reinhardtii* (EMPIAR-10694). Our results (presented in the new Figure 6 and Suppl. Figure 5) demonstrate MiLoPYP's ability to successfully detect and localize particles of ATP synthase complexes from EMPIAR-11658 and crowded RuBisCo enzymes (~500kDa complex) from EMPIAR-10694. Together, these results more convincingly support the validity of our model.

Comment 4. In the Online Methods section, the author uses several random enhancement in the contrastive learning module to maintain instance identity and promote effective learning. But this usually makes the convergence of the model worse. Therefore, we hope that the authors will demonstrate the necessity of incorporating these training methods through ablation experiments.

It is possible that the reviewer missed this, but ablation experiments for the contrastive learning module were included in the original Supplementary Figure 3. As demonstrated in the original SimCLR paper (<https://doi.org/10.48550/arXiv.2002.05709>) and later in the BYOL (<https://arxiv.org/abs/2006.07733>) and SwAV (<https://arxiv.org/abs/2006.09882>) papers, data augmentation is crucial to yield effective representations when using contrastive learning, and if removed, it can result in 5-25% decrease in accuracy. Since MiLoPYP performs unsupervised contrastive learning without negative pairs, removing any of its data augmentation strategies will fail to produce meaningful outputs and result in model collapse. In this case, while the model may seem to "converge" and result in low loss values, the model is actually producing the same meaningless representation for all input subvolumes.

Comment 5. In the section 'Accurate detection and localization of proteins imaged in vitro', the author compares the proposed model with conventional template matching and the deep learning-based approach crYOLO-3D. However, the author did not give the parameter settings used when running the comparison method. The effect of deep learning methods often depends on the parameters used, so we hope that the authors can add explanations of the parameter values in the two comparison methods to show the fairness of the comparison.

When doing comparisons with other methods, we always followed the recommended procedures and selected parameters that gave the best possible results. For template-matching on EMPIAR-10304, we used the EMAN2 command `e2spt_tmpmatch.py` with the following parameters: `nptcl=1200, dthr=10, vthr=2, delta=30.0, sym=c1, rmedge=true boxsz=-1`. For EMPIAR-10499, we used template-matching as implemented in the package Warp. First, we manually picked particles that were used for generating an initial reference (pixel size=10 Å, box size=34). This reference was low-pass filtered to 40 Å and used for template matching in Warp. Particles were selected at positions that had `_rlnAutopickFigureOfMerit > 13`.

For crYOLO-3D, we retrained the model on each dataset using manually picked particles across 10 consecutive slices (selecting 20-30% of particles) and using tomograms with different defocus values. We first created a common configuration file using the command:

```
cryolo_gui.py --ignore-goey config --a crYOLO -nm STANDARD --num_patches 1
--overlap_patches 200 --filtered_output filtered_tmp -f LOWPASS --low_pass_cutoff 0.4
--janni_overlap 24 --janni_batches 3 --train_times 10 --batch_size 4 --learning_rate 0.0001
--nb_epoch 200 --object_scale 5.0 --no_object_scale 1.0 --coord_scale 1.0 --class_scale 1.0
{box size},
```

where {box size} was set to 16 for EMPIAR-10499 and to 12 for EMPIAR-10304. The following command was used for training:

```
cryolo_gui.py --ignore-goey train -c config_cryolo.json -w 5 -nc -1 --gpu_fraction 1.0 -e 10 -lft 2
--seed 10.
```

For the prediction stage, we used the following commands:

```
● EMPIAR-10499: cryolo_gui.py predict -c config_cryolo.json -w cryolo_model.h5 -i
TS_01_rec -o cryolo_output -t 0.102 -pbs 3 --gpu_fraction 1.0 -nc -1 -mw 100 -sr 1.41
--tomogram -tsr 20 -tmem 18 -tmin 6.
```

```
● EMPIAR-10304: cryolo_gui.py predict -c config_cryolo.json -w cryolo_model.h5 -i
TS_01_rec -o cryolo_output -t 0.102 -pbs 3 --gpu_fraction 1.0 -nc -1 -mw 100 -sr 1.41
--tomogram -tsr 16 -tmem 14 -tmin 6.
```

We added some of these details to the Online Methods section.

Comment 6. There is a spelling error in "constrastive" on line 58, it should be "contrastive". For "time consuming" on line 64, a hyphen needs to be added between "time" and "consuming". "STA" on line 99 may need to be explained or given its full name to ensure that readers understand it.

Thanks for noticing these, we corrected all of them in the revision.

Comment 7. In the discussion section, the author mentioned that the 2D visualization step in MiLoPYP is prone to class imbalance. Please analyze this quantitatively. For example, the authors could show changes in model performance when changing the number of neighbors in UMAP and tsne low-dimensional embeddings.

It is worth noting that the determination of classes in the UMAP and t-SNE embeddings is only done for the purpose of visualization and it does not affect the selection of particles used to train the refinement module. Nevertheless, following the reviewer's suggestion, we ran experiments using 10, 40 and 80 neighbors to show the effect of this parameter on the analysis of three datasets: EMPIAR-10304, EMPIAR-10499 and EMPIAR-10987 (new Supplementary Figure 8). The use of fewer neighbors reveals finer local structure while the use of more neighbors captures the overall structure with some loss of fine local structure. However, regardless of the number of neighbors, the full vector representation for each subvolume remains unchanged as UMAP only serves as a post-processing dimensionality reduction technique for the purpose of visualization. We clarified these points in the Online Methods.

Version 2:

Decision Letter:

Our ref: NMETH-A54834B

23rd May 2024

Dear Alberto,

Thank you for submitting your revised manuscript "MiLoPYP: self-supervised molecular pattern mining and particle localization in situ" (NMETH-A54834B). It has now been seen by the original referees and their comments are below. The reviewers find that the paper has improved in revision, and therefore we'll be happy in principle to publish it in Nature Methods, pending minor revisions to satisfy the referees' final requests (as discussed) and to comply with our editorial and formatting guidelines.

Please note, if for some reason EMDB will not accept your additional maps, we ask that you share them via e.g. Figshare.

TRANSPARENT PEER REVIEW

ORCID

IMPORTANT: Non-corresponding authors do not have to link their ORCID but are encouraged to do so. Please note that it will not be possible to add/modify ORCIDs at proof. Thus, please let your co-authors know that if they wish to have their ORCID added to the paper they must follow the procedure described in the following link prior to acceptance: <https://www.springernature.com/gp/researchers/orcid/orcid-for-nature-research>

Sincerely,
Rita

Rita Strack, Ph.D.
Senior Editor
Nature Methods

Reviewer #1 (Remarks to the Author):

The revised manuscript on the MiLoPYP pipeline for cellular mining and particle localisation for subtomogram analysis incorporates a number of changes in response to the previous reviewer comments. Most notably, cellular applications of the pipeline have been added to the manuscript, such as the identification of RuBisCo particles from *C. reinhardtii* and ATP synthase in *S. cerevisiae* mitochondria. The authors addressed the reviewer comments in a rigorous manner and softened claims where necessary. The interesting new ideas in MiLoPYP combined with the convenient usage of the package will possibly enable new observations within existing datasets and generally simplify steps in the analysis of cryo-ET data.

In conclusion, I support the publication of the manuscript in Nature Methods. I believe it is going to be a valuable addition to the tomographer's tool belt. Below only a few comments on the revised manuscript.

1. Regarding the data availability statement, I encourage the authors to upload all obtained maps to the EMDB such that readers can make their own judgement on the map qualities from the MiLoPYP + NextPYP pipeline. Currently, these are only provided to the reviewers in private. RuBisCo and ATP synthase reconstructions were missing in this data repository.
2. A minor comment but there is likely a mistake/typo in Supplementary Table 3 for the timing of few shot particle localisation as both Training (10 epochs) and inference are assigned a timing of 0:03:25 (hh:mm:ss). It would be rather surprising, though not impossible, for these two times to be the same.

Reviewer #1 (Remarks on figshare data availability):

Densities for the RuBisCo and ATP synthase subtomogram averages were missing.

Reviewer #2 (Remarks to the Author):

My questions have been well addressed in the revised manuscript. I have one more suggestion. Is it possible to give a rough estimation for the recall of the particle recognition? While a ground truth is missing, a rough estimation based on visual inspection should be enough. The recall is important for estimating the performance of the picking and let the user know how many particles are not picked in a crowded cellular environment.

Reviewer #3 (Remarks to the Author):

The authors have answered all of my last questions and concerns, so I think this version is good.

Author Rebuttal letter:

We thank the reviewers for evaluating our work and for the constructive feedback. Our point-to-point responses are included below (highlighted in blue).

Reviewers' Comments:

Reviewer #1:

Remarks to the Author:

The revised manuscript on the MiLoPYP pipeline for cellular mining and particle localisation for subtomogram analysis incorporates a number of changes in response to the previous reviewer comments. Most notably, cellular applications of the pipeline have been added to the manuscript, such as the identification of RuBisCo particles from *C. reinhardtii* and ATP synthase in *S. cerevisiae* mitochondria. The authors addressed the reviewer comments in a rigorous manner and softened claims where necessary. The interesting new ideas in MiLoPYP combined with the convenient usage of the package will possibly enable new observations within existing datasets and generally simplify steps in the analysis of cryo-ET data.

In conclusion, I support the publication of the manuscript in Nature Methods. I believe it is going to be a valuable addition to the tomographer's tool belt. Below only a few comments on the revised manuscript.

1. Regarding the data availability statement, I encourage the authors to upload all obtained maps to the EMDB such that readers can make their own judgement on the map qualities from the MiLoPYP + NextPYP pipeline. Currently, these are only provided to the reviewers in private. RuBisCo and ATP synthase reconstructions were missing in this data repository.

We deposited all the maps in the EMDB including the ones for RuBisCo and ATP synthase. We provide the accession codes in the Data Availability section.

2. A minor comment but there is likely a mistake/typo in Supplementary Table 3 for the timing of few shot particle localisation as both Training (10 epochs) and inference are assigned a timing of 0:03:25 (hh:mm:ss). It would be rather surprising, though not impossible, for these two times to be the same.

Thank you for catching this. The correct timing for inference should have been 0:10:26 (hh:mm:ss). This corresponds to all tomograms in the dataset.

Reviewer #2:

Remarks to the Author:

My questions have been well addressed in the revised manuscript. I have one more suggestion. Is it possible to give a rough estimation for the recall of the particle recognition? While a ground truth is missing, a rough estimation based on visual inspection should be enough. The recall is important for estimating the performance of the picking and let the user know how many particles are not picked in a crowded cellular environment.

Maybe the reviewer missed this, but we reported recall values in the original Supplementary Table 2 for the EMPIAR datasets: 10304, 10453, 10499, and 10987. Recall values range between 0.69 and 0.86 depending on the dataset.

Reviewer #3:

Remarks to the Author:

The authors have answered all of my last questions and concerns, so I think this version is good.

We thank the reviewer for the positive feedback.

Version 3:

Decision Letter:

5th Aug 2024

Dear Alberto,

I am pleased to inform you that your Article, "MiLoPYP: self-supervised molecular pattern mining and particle localization in situ", has now been accepted for publication in Nature Methods. The received and accepted dates will be Dec 26, 2023 and August 5, 2024. This note is intended to let you know what to expect from us over the next month or so, and to let you know where to address any further questions.

Over the next few weeks, your paper will be copyedited to ensure that it conforms to Nature Methods style. Once your paper is typeset, you will receive an email with a link to choose the appropriate publishing options for your paper and our Author Services team will be in touch regarding any additional information that may be required. It is extremely important that you let us know now whether you will be difficult to contact over the next month. If this is the case, we ask that you send us the contact information (email, phone and fax) of someone who will be able to check the proofs and deal with any last-minute problems.

Please note that *Nature Methods* is a Transformative Journal (TJ). Authors may publish their research with us through the traditional subscription access route or make their paper immediately open access through payment of an article-processing charge (APC). Authors will not be required to make a final decision about access to their article until it has been accepted. [Find out more about Transformative Journals](https://www.springernature.com/gp/open-research/transformative-journals)

If you have any questions about our publishing options, costs, Open Access requirements, or our legal forms, please contact

ASJournals@springernature.com

If you are active on Twitter/X, please e-mail me your and your coauthors' handles so that we may tag you when the paper is published.

Best regards,
Rita

Rita Strack, Ph.D.
Senior Editor
Nature Methods

Visit the Springer Nature Editorial and Publishing website at http://editorial-jobs.springernature.com?utm_source=ejP_NMeth_email&utm_medium=ejP_NMeth_email&utm_campaign=ejp_Nmeth for more information about our career opportunities. If you have any questions please click [here](mailto:editorial.publishing.jobs@springernature.com).
